# Repurposing the mammalian RNA-binding protein Musashi-1 as an allosteric translation repressor in bacteria

**Roswitha Dolcemascolo**[1,2†], **María Heras-Hernández**[1†], **Lucas Goiriz**[1,3†], **Roser Montagud-Martínez**[1,2], **Alejandro Requena-Menéndez**[1], **Raúl Ruiz**[1], **Anna Pérez-Ràfols**[4,5], **R Anahí Higuera-Rodríguez**[6,7], **Guillermo Pérez-Ropero**[8,9], **Wim F Vranken**[10,11], **Tommaso Martelli**[4], **Wolfgang Kaiser**[6], **Jos Buijs**[8,12], **Guillermo Rodrigo**[1*]

[1]Institute for Integrative Systems Biology (I2SysBio), CSIC – University of Valencia, Paterna, Spain; [2]Department of Biotechnology, Polytechnic University of Valencia, Valencia, Spain; [3]Department of Applied Mathematics, Polytechnic University of Valencia, Valencia, Spain; [4]Giotto Biotech SRL, Sesto Fiorentino, Italy; [5]Magnetic Resonance Center (CERM), Department of Chemistry Ugo Schiff, Consorzio Interuniversitario Risonanze Magnetiche di Metalloproteine (CIRMMP), University of Florence, Sesto Fiorentino, Italy; [6]Dynamic Biosensors GmbH, Planegg, Germany; [7]Department of Physics, Technical University of Munich, Garching, Germany; [8]Ridgeview Instruments AB, Uppsala, Sweden; [9]Department of Chemistry – BMC, Uppsala University, Uppsala, Sweden; [10]Structural Biology Brussels, Vrije Universiteit Brussel, Brussels, Belgium; [11]Interuniversity Institute of Bioinformatics in Brussels, Université Libre de Bruxelles – Vrije Universiteit Brussel, Brussels, Belgium; [12]Department of Immunology, Genetics, and Pathology, Uppsala University, Uppsala, Sweden

**\*For correspondence:**
guillermo.rodrigo@csic.es

[†]These authors contributed equally to this work

**Abstract** The RNA recognition motif (RRM) is the most common RNA-binding protein domain identified in nature. However, RRM-containing proteins are only prevalent in eukaryotic phyla, in which they play central regulatory roles. Here, we engineered an orthogonal post-transcriptional control system of gene expression in the bacterium *Escherichia coli* with the mammalian RNA-binding protein Musashi-1, which is a stem cell marker with neurodevelopmental role that contains two canonical RRMs. In the circuit, Musashi-1 is regulated transcriptionally and works as an allosteric translation repressor thanks to a specific interaction with the N-terminal coding region of a messenger RNA and its structural plasticity to respond to fatty acids. We fully characterized the genetic system at the population and single-cell levels showing a significant fold change in reporter expression, and the underlying molecular mechanism by assessing the in vitro binding kinetics and in vivo functionality of a series of RNA mutants. The dynamic response of the system was well recapitulated by a bottom-up mathematical model. Moreover, we applied the post-transcriptional mechanism engineered with Musashi-1 to specifically regulate a gene within an operon, implement combinatorial regulation, and reduce protein expression noise. This work illustrates how RRM-based regulation can be adapted to simple organisms, thereby adding a new regulatory layer in prokaryotes for translation control.

## eLife assessment

This **important** study demonstrates the use of the mammalian Musashi-1 (MSI-1) RNA-binding protein as a tool for regulating gene expression in *Escherichia coli*. The authors provide **convincing** evidence that MSI-1 functions as an effective repressor of translation, and that MSI-1 can be allosterically controlled by oleic acid. This work establishes MSI-1 as a potential tool for synthetic biology applications, and the system developed here can be used for mechanistic studies of MSI-1.

## Introduction

Gene regulation at the post-transcriptional level is pervasive in living organisms of ranging complexity (*Waters and Storz, 2009*; *Holmqvist and Vogel, 2018*; *Jonas and Izaurralde, 2015*; *Glisovic et al., 2008*). Indeed, the ability to regulate the genetic information flow at different points appears instrumental to maximize the integration of intrinsic and extrinsic signals, which enables an efficient information processing by the organisms. However, the solutions implemented in prokaryotes and eukaryotes greatly differ. In prokaryotes, small RNAs (sRNAs) regulate messenger RNA (mRNA) stability and translation initiation (*Waters and Storz, 2009*), supported by a series of RNA-binding proteins (e.g., Hfq) that act globally (*Holmqvist and Vogel, 2018*). Regulatory proteins of specific scope in these simple organisms mainly operate in the transcriptional layer (*Madan Babu et al., 2006*), what is aligned with the models presented in the early times of molecular biology (*Jacob and Monod, 1961*). By contrast, eukaryotes deploy a sizeable number of RNA-binding proteins with a variety of functions (*Glisovic et al., 2008*) that participate in the regulation of mRNA turnover, transport, splicing, and translation in a gene-specific manner and also at a global scale. In animals, in particular, most RNA-binding proteins contain RNA recognition motifs (RRMs) (*Maris et al., 2005*). RRMs are small globular domains of about 90 amino acids that fold into four antiparallel β-strands and two α-helices, which can bind to single-strand RNAs with sufficient affinity and specificity to control biological processes (*Messias and Sattler, 2004*). Yet, while important to attain functional diversity in the post-transcriptional layer in animals, RRMs are not prevalent in all organisms. In fact, the scarcity of RRM-containing proteins in prokaryotes and the often-unknown functional role of those identified by bioinformatic methods (*Maruyama et al., 1999*) question whether RRMs can readily work in organisms with much simpler gene expression machinery and intracellular organization. If so, this would raise the potential to use RRM–RNA interactions as an orthogonal layer to engineer gene regulation in prokaryotes.

To address these intriguing questions, we adopted a synthetic biology approach where a specific RRM-containing protein was incorporated in a bacterium in order to engineer a post-transcriptional control module. Synthetic biology has highlighted how living cells can be (re)programmed through the assembly of independent genetic elements into functional networks for a variety of applications in biotechnology and biomedicine (*Khalil and Collins, 2010*). Yet, synthetic biology can also be used to disentangle natural systems and probe hypotheses about biological function (*Bashor and Collins, 2018*). In previous work, some proteins with the ability to recognize RNA have been exploited as translation factors in bacteria for a gene-specific regulation (*Belmont and Niles, 2010*; *Katz et al., 2019*; *Cao et al., 2015*). The first instance was the tetracycline repressor protein (TetR), which naturally functions as a transcription factor, by means of the selection of synthetic RNA aptamers (*Belmont and Niles, 2010*). The bacteriophage MS2 coat protein (MS2CP) (*Katz et al., 2019*) and eukaryotic Pumilio homology domains (*Cao et al., 2015*) were also used in synthetic circuits. Alternatively, a wide palette of post-transcriptional control systems based on sRNAs have been developed in recent years to program gene expression in bacteria (*Qi and Arkin, 2014*). Of note, these systems are amenable to be combined with regulatory proteins to attain complex dynamic behaviors (*Rosado et al., 2018*). A heterologous RRM-containing protein with definite regulatory activity, in addition to provide empirical evidence on the adaptability of such RNA-binding domains to different genetic backgrounds, would enlarge the synthetic biology toolkit (*Shotwell et al., 2020*), boosting applications in which high orthogonality, expression fine-tuning, and signal integrability are required features. In addition, RRMs can themselves be allosterically regulated, opening up new avenues for post-transcriptional regulation by small molecules.

Still, there are instances of bacterial proteins that regulate translation in a gene-specific manner, such as CsrA to control glycogen biosynthesis (*Liu and Romeo, 1997*) or the ribosomal protein S8 to exert self-repression (*Meyer, 2018*). Besides, it is worth noting that some bacteriophages follow this mechanism to modulate their infection cycle. These are the cases, for example, of the coat proteins

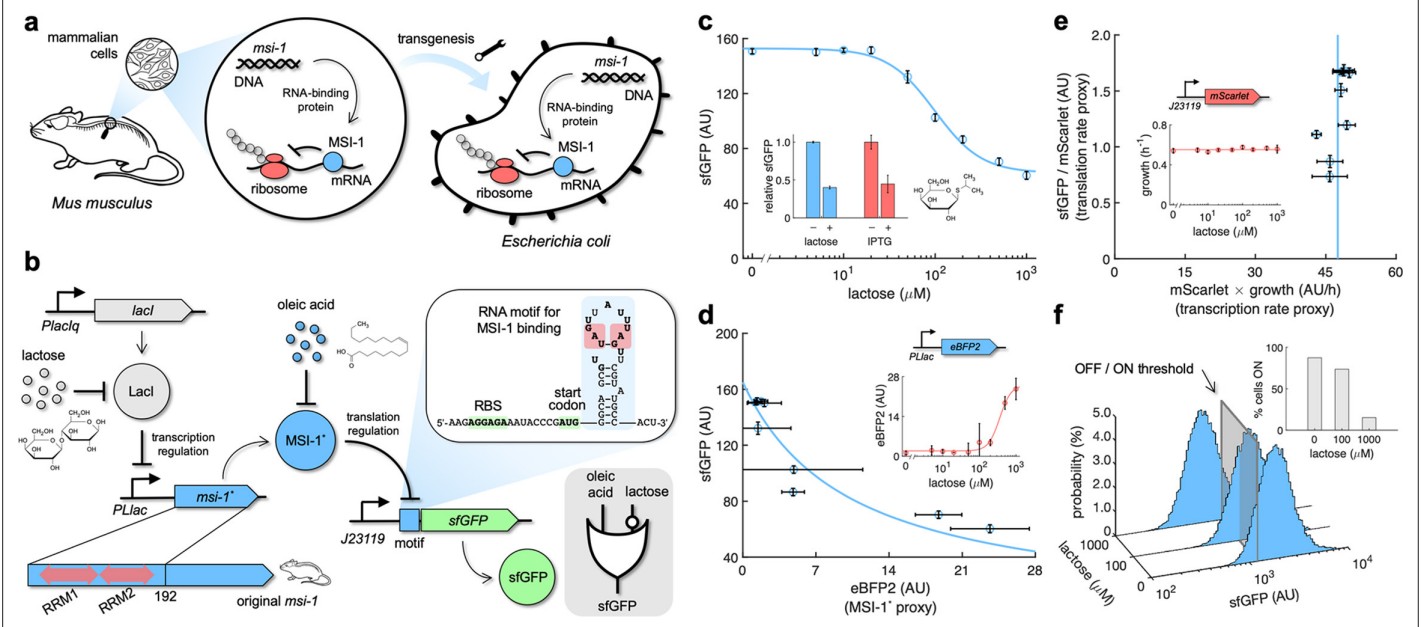

**Figure 1.** Musashi-1 can downregulate translation in bacteria. (**a**) Overview of the biotechnological development. In mammals, MSI-1 binds to the 3' untranslated region (UTR) of its target mRNA to repress translation. Here, the *M. musculus* gene coding for MSI-1 was moved to *E. coli* (transgenesis) to implement a synthetic regulation system at the level of translation. (**b**) Schematic of the synthetic gene circuit engineered in *E. coli*. A truncated version of MSI-1 (termed MSI-1*) was expressed from the PLlac promoter to be induced with lactose (or isopropyl β-D-1-thiogalactopyranoside [IPTG]) in a genetic background overexpressing LacI. sfGFP was used as a reporter expressed from a constitutive promoter (J23119) and under the control of a suitable RNA motif recognized by MSI-1* in the N-terminal coding region of the transcript (viz., located after the start codon). The activity of MSI-1* could in turn be allosterically inhibited by oleic acid. In electronic terms, this circuit implements an IMPLY logic gate. The inset shows the predicted secondary structure of the N-terminal coding region of the reporter mRNA. Within the motif (blue shaded), the consensus recognition sequences (RU$_n$AGU) are bolded and the minimal cores (UAG) are marked in red. System implemented with pRM1+ and pREP6. (**c**) Dose–response curve of the system using lactose as inducer (up to 1 mM). MSI-1* downregulated sfGFP expression by 2.5-fold. The inset shows the dynamic range of the response using lactose or IPTG (1 mM), showing a statistically significant regulation in both cases (Welch's *t*-test, two-tailed p<0.05). (**d**) Transfer function of the system (between sfGFP and MSI-1*). The inset shows the dose–response curve of eBFP2 expressed from the PLlac promoter (proxy of MSI-1* expression) with lactose. (**e**) Scatter plot of the dynamic response of the system in the Crick space (translation rate vs. transcription rate). The dose–response curve of mScarlet expressed from the J23119 promoter with lactose was used to perform the decomposition (vertical line fitted to 48 AU/h). The inset shows the growth rate of the cells for each induction condition (horizontal line fitted to 0.55 h$^{-1}$). In all cases, points correspond to experimental data, while solid lines come from adjusted mathematical models. Error bars correspond to standard deviations (n = 3). (**f**) Probability-based histograms of sfGFP expression from single-cell data for different lactose concentrations, showing a statistically significant regulation (one-way ANOVA, p<10$^{-4}$). The inset shows the percentage of cells in the ON state (sfGFP expressed), according to a specified threshold, for each lactose concentration. AU, arbitrary units.

The online version of this article includes the following source data and figure supplement(s) for figure 1:

**Source data 1.** Bulk fluorescence data of sfGFP, eBFP2, and mScarlet with lactose and single-cell data of sfGFP.

**Figure supplement 1.** Maps of the plasmids used to implement the synthetic gene circuit in which MSI-1* represses the translation of sfGFP.

**Figure supplement 2.** RT-qPCR results for the mRNA level of *sfGFP*.

of the phages MS2 (infecting *Escherichia coli*) or PP7 (infecting *Pseudomonas aeruginosa*), which regulate the expression of the cognate phage replicases through protein–RNA interactions (*Babitzke et al., 2009*). However, one limitation for synthetic biology developments is that such phage proteins are not allosteric. At the post-transcriptional level, bacteria mostly rely on a large palette of *cis*- and *trans*-acting non-coding RNAs to either activate or repress protein expression, resulting in the regulation of translation initiation, mRNA stability, or transcription termination, and even allowing sensing small molecules (*Waters and Storz, 2009*; *Qi and Arkin, 2014*). Thus, there should be efforts to replicate this functional versatility with proteins.

In this work, the mammalian RNA-binding protein Musashi-1 (MSI-1) (*Fox et al., 2015*) was used as a translation repressor in the bacterium *E. coli* (*Figure 1a*). MSI-1 belongs to an evolutionarily conserved family of RRM-containing proteins, of which a member was first identified in *Drosophila melanogaster*

(*Nakamura et al., 1994*). MSI-1 contains two RRMs in the N-terminal region (RRM1 and RRM2) and recognizes the RNA consensus sequence RU$_n$AGU on the nanomolar affinity scale (*Imai et al., 2001*). Importantly, MSI-1 can be allosterically inhibited by fatty acids (in particular, 18–22-carbon $\omega$–9 mono-unsaturated fatty acids) (*Clingman et al., 2014*). In mammals, MSI-1 is mainly expressed in stem cells of neural and epithelial lineage and plays crucial roles in differentiation, tumorigenesis, and cell cycle regulation (*Fox et al., 2015*). Notably, MSI-1 regulates Notch signaling by repressing the translation of a key protein in the pathway (*Imai et al., 2001*). Hence, rather than moving genetic elements from simple to complex organisms, as it is normally done (e.g., the TetR-aptamer module was implemented in simple eukaryotes *Ganesan et al., 2016*), we reversed the path by moving an important mamma-lian gene (from *Mus musculus*) to *E. coli*. Some eukaryotic factors have already been implemented in bacteria to regulate gene expression at different levels (*Cao et al., 2015*; *MacDonald et al., 2021*), but the case of RRM-containing proteins has remained elusive. In the following, we present quantita-tive experimental and theoretical results on the response dynamics of a synthetic gene circuit in which MSI-1 works as an allosteric translation repressor. There, MSI-1 is transcriptionally controlled by the lactose repressor protein (LacI), and translation regulation by MSI-1 is accomplished by means of a specific interaction with an mRNA (encoding a reporter protein) that harbors a suitable binding motif in its N-terminal coding region.

## Results

### A Musashi protein can downregulate translation in bacteria

From the amino acid sequence of *M. musculus* MSI-1, we generated a nucleotide sequence with codons optimized for *E. coli* expression. Knowing that the C-terminus of MSI-1 is of low structural complexity (*Iwaoka et al., 2017*), we cloned a truncated version of the gene encompassing the first 192 amino acids, which include the two RRMs, to implement our synthetic circuit (*Figure 1b*). The resulting protein (termed MSI-1*) was expressed from a synthetic PL-based promoter repressed by LacI (termed PLlac) (*Lutz and Bujard, 1997*) lying in a high copy number plasmid. This allowed controlling the expression of the heterologous RNA-binding protein at the transcriptional level with lactose or isopropyl β-D-1-thiogalactopyranoside (IPTG) in a genetic background overexpressing LacI. As a regulated element, we used the superfolder green fluorescent protein (sfGFP) (*Pédelacq et al., 2006*), which was expressed from a constitutive promoter (J23119) lying in a low copy number plasmid (*Figure 1—figure supplement 1*). An RNA motif obtained by affinity elution-based RNA selection (SELEX) containing two copies of the consensus recognition sequence (viz., GUUAGU and AUUUAGU) (*Imai et al., 2001*) was placed in frame after the start codon of *sfGFP*. This motif folds into a stem-loop structure that allows stabilizing the exposure of the recognition sequence to the solvent. In this way, MSI-1* can repress translation by blocking the binding of the ribosome, presumably by imposing a steric hindrance for the 30S ribosomal subunit. This mode of action differs from the natural one in mammals, in which MSI-1 binds to the 3′ untranslated region (UTR) of its target mRNA (*Numb*) to repress translation by disrupting the activation function of the poly(A)-binding protein (*Kawahara et al., 2008*). Here, considering lactose (or IPTG) and oleic acid as the two inputs and sfGFP as the output, MSI-1* being an internal allosteric regulator operating at the post-transcriptional level, an IMPLY gate would model the logic behavior of the resulting circuit (i.e., sfGFP would only turn off with lactose and without oleic acid in the medium).

We first characterized by bulk fluorometry the dose–response curve of the system using a lactose concentration gradient up to 1 mM. Our data show that MSI-1* downregulated sfGFP expression by 2.5-fold (*Figure 1c*). Fitting a Hill equation, we obtained a regulatory coefficient of 99 μM (lactose concentration at which the repression is half of the maximal) and a Hill coefficient of 1.7 (Appendix 1). We also observed that IPTG (a synthetic compound) triggered a very similar response. To further inspect the activity of the RNA-binding protein, we filtered out the transcriptional regulatory effect. For that, we expressed the enhanced blue fluorescent protein 2 (eBFP2) (*Ai et al., 2007*) from the PLlac promoter to obtain the corresponding dose–response curve with lactose. In this way, eBFP2 expression was a proxy of MSI-1* expression, which allowed representing the transfer function of the engineered regulation (*Figure 1d*). A Hill equation with no cooperative binding (i.e., Hill coefficient of 1) explained the data with sufficient agreement, suggesting that only one protein interacted with a given mRNA (i.e., each RRM of MSI-1* binds to a consensus sequence repeat, in agreement with a

previous structural model; *Iwaoka et al., 2017*). We also measured the cell growth rate for all induction conditions, finding that the values were almost constant. This indicates that the expression of the mammalian protein did not produce a significant burden to the bacterial cell.

In simple terms, protein expression comes from the product of the transcription and translation rates of the gene. Hence, we examined such a decomposition in the case of sfGFP expression regulated by MSI-1*. Of note, the low copy number plasmid harbors an additional transcriptional unit to express the monomeric red fluorescent protein mScarlet (*Bindels et al., 2017*) from a constitutive promoter (J23119). We then monitored its expression profile with lactose. Assuming that *sfGFP* and *mScarlet* were equally transcribed, as they were expressed from the same promoter, and that the translation rate of *mScarlet* was constant, the product of mScarlet expression and cell growth rate was considered a proxy of the transcription rate of *sfGFP*. Moreover, the ratio of sfGFP and mScarlet expressions was a proxy of the translation rate of *sfGFP* (*Klumpp et al., 2009*). This served us to represent the dynamics of the system in a plane defined as translation rate vs. transcription rate (termed Crick space; *Hausser et al., 2019*), highlighting that the change in sfGFP expression with lactose comes indeed from translation regulation (*Figure 1e*). A reverse transcription quantitative polymerase chain reaction (RT-qPCR) was used to confirm the preservation of the *sfGFP* mRNA level (*Figure 1—figure supplement 2*). Finally, to evaluate the heterogeneity of the response within a bacterial population, we performed single-cell measurements of sfGFP expression by flow cytometry. Unimodal distributions able to shift in response to lactose were observed (*Figure 1f*). Setting a threshold to categorize expression, we found that the percentage of cells in the ON state dropped from 87% to 15% upon addition of 1 mM lactose. In sum, our results show that MSI-1* can regulate translation in a specific manner in *E. coli*, and hence that eukaryotic regulators can be borrowed to be functional elements in prokaryotes.

## Mechanistic insight into the engineered regulation based on a protein–RNA interaction

We then introduced a series of point mutations into the SELEX RNA motif to assess their effect over the regulatory activity of the RRM-containing protein (*Figure 2a*). These mutations change the consensus recognition sequence of at least one repeat. A characterization of all systems revealed that the mutations affected both the maximal level and fold change of sfGFP expression (*Figure 2b*). Of note, a single point mutation in one repeat leading to RU$_n$CGU (mutant 1) was quite detrimental for the MSI-1*-based regulation (only 1.4-fold reduction in sfGFP expression). Despite the mutation substantially reducing sfGFP expression in the absence of MSI-1*, the presumed repressed state upon addition of lactose did not change much, suggesting the difficulty of the protein for targeting the mutated mRNA. This agrees with the prior observation that, within the consensus sequence, UAG is a minimal core that determines the specific recognition by MSI-1 (*Zearfoss et al., 2014*). A double point mutation changing the minimal cores of the two repeats (UAC rather than UAG; mutant 5) also resulted in a detrimental action, but not to a greater extent. We also engineered a new reporter system with a minimal RNA motif consisting of a single copy of the shortest possible consensus sequence (AUAGU), but its characterization showed no apparent regulation by MSI-1* (*Figure 2—figure supplement 1*). Taken together, two copies of the consensus sequence seem necessary for a successful regulation of protein expression.

To relate the cellular effects with protein–RNA interactions, we obtained a purified MSI-1* preparation in order to perform in vitro binding kinetics assays (*Figure 2—figure supplement 2*). For that, a gene coding for a truncated version of the human MSI-1 was expressed from a T7 polymerase promoter in *E. coli*. With respect to the *M. musculus* version, this protein only differs in one residue of RRM2 (then termed MSI-1$_h$*), which is the subsidiary domain for RNA recognition (note also that the human and mouse proteins recognize the same consensus sequence; *Zearfoss et al., 2014*). To avoid the necessity of labeling the molecules of interest and allow working with very low amounts of protein and RNA, we used the switchSENSE technology, which allows measuring molecular dynamics on a chip (*Figure 2c*; *Cléry et al., 2017*). *Figure 2d* summarizes the resulting protein–RNA association and dissociation rates ($k_{ON}$ and $k_{OFF}$, respectively; see also *Figure 2—figure supplement 3*). In the case of the original RNA motif, we found an association rate of 1.1 nM$^{-1}$ min$^{-1}$, which means that a single regulator molecule would take 1–3 min to find its target in the cell, and a residence time of the protein on the RNA of 1.5 min (given by $1/k_{OFF}$). Of note, the reported value of $k_{ON}$ is relatively close to the upper

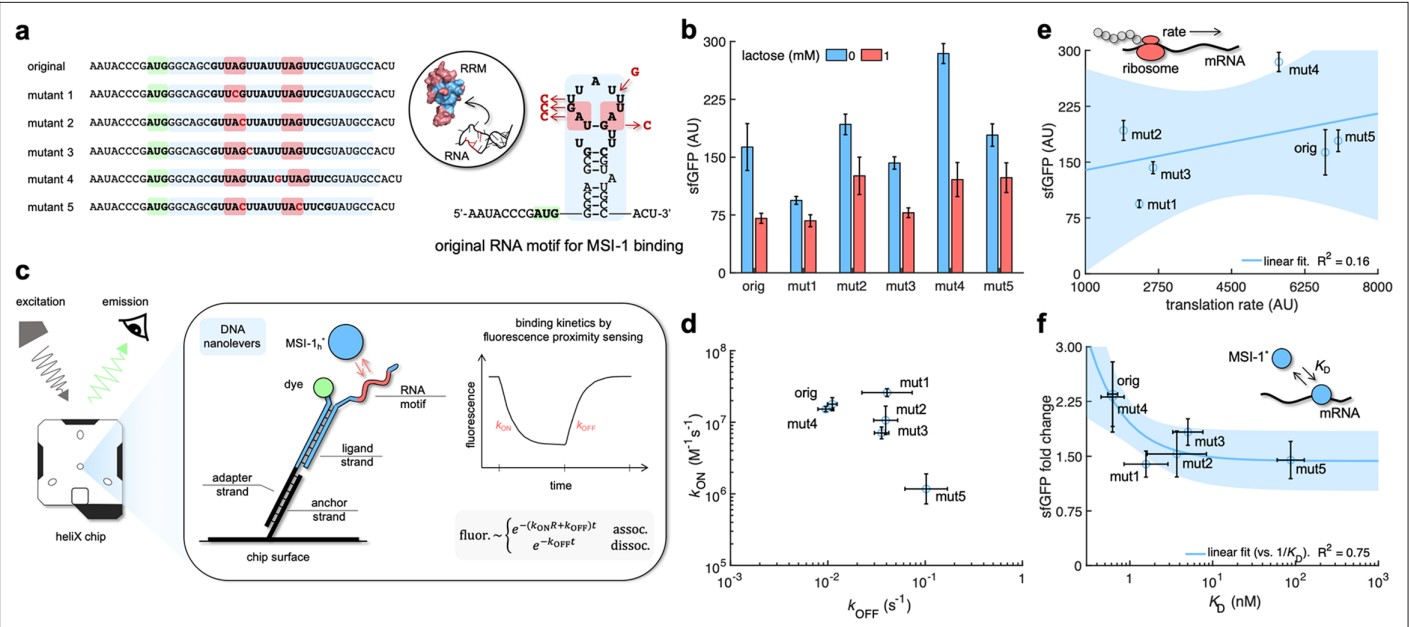

**Figure 2.** Mechanistic characterization of the Musashi-1–mRNA interaction. (**a**) Sequences and predicted secondary structures of the different RNA motif variants for MSI-1 binding analyzed in this work. Point-mutations indicated in red. Three-dimensional representations of the RRM1 and RNA motif are also shown. Within the RRM1, the region that recognizes the RNA is shown in blue. (**b**) Dynamic range of the response of the different genetic systems using lactose (1 mM), showing a statistically significant regulation in all cases (Welch's $t$-test, two-tailed $p < 0.05$; although some mutants present a small fold change). (**c**) Schematic of the heliX biosensor platform. A double-strand DNA nanolever was immobilized on a gold electrode of the chip. The nanolever carried a fluorophore in one end and the RNA motif for MSI-1 binding in the other. Binding between MSI-1$_h$* (injected analyte) and RNA led to a fluorescence change, whose monitoring in real time served to extract the kinetic constants that characterize the interaction. (**d**) Scatter plot of the experimentally-determined kinetic constants of association and dissociation between the protein and the RNA for all systems (original and five mutants). Means and deviations calculated in log scale (geometric). (**e**) Correlation between the maximal sfGFP expression level (in the absence of lactose) and the translation rate predicted with RBS calculator. Linear regression performed. (**f**) Correlation between the fold change in sfGFP expression and the dissociation constant ($K_D$). Deviations calculated by propagation. Linear regression performed (vs. $1/K_D$). Blue shaded areas indicate 95% confidence intervals. In all cases, error bars correspond to standard deviations (n = 3). AU, arbitrary units.

The online version of this article includes the following source data and figure supplement(s) for figure 2:

**Source data 1.** Bulk fluorescence data of sfGFP and binding kinetics measurements.

**Figure supplement 1.** Characterization of the system response with lactose using pREP4 as a reporter plasmid.

**Figure supplement 2.** Musashi protein purification.

**Figure supplement 3.** Characterization of different mutant RNA motifs in terms of binding kinetics against the MSI-1$_h$* protein.

limit imposed by the diffusion rate (~1 nM$^{-1}$ s$^{-1}$). This fast rate suggests that MSI-1* is able to find its target mRNA in *E. coli*, competing with ribosomes and ribonucleases, and then achieve translation regulation. We also found that a single mutation in one of the two UAG minimal cores (mutants 1 and 2) led to similar association but faster dissociation (almost four times faster dissociation), whereas a double mutation affecting the two cores (mutant 5) disturbed both phases (almost 15 times slower association and 10 times faster dissociation). The dissociation constant ($K_D = k_{OFF}/k_{ON}$) was 0.62 nM for the original system, while 87 nM for mutant 5. The switchSENSE technology allowed revealing that affinity on the subnanomolar scale, refining a previous estimate of 4 nM obtained by gel shift assays (*Imai et al., 2001*). To contextualize these values, we compared to the binding kinetics of MS2CP, a phage RNA-binding protein that has evolved in a prokaryotic context and that we recently exploited to study how expression noise emerges and propagates through translation regulation (*Dolcemascolo et al., 2022*). Previous work disclosed an association rate to the cognate RNA motif of 0.032 nM$^{-1}$ min$^{-1}$ and a residence time of 12 min, leading to a dissociation constant of 2.6 nM (*Buenrostro et al., 2014*). Thus, MSI-1* would target RNA faster than MS2CP, but once this happened the phage protein would remain bound longer.

Next, we tried to predict the impact of the mutations on sfGFP expression. On the one hand, we used an empirical free-energy model (RBS calculator) to obtain an estimate of the mRNA translation

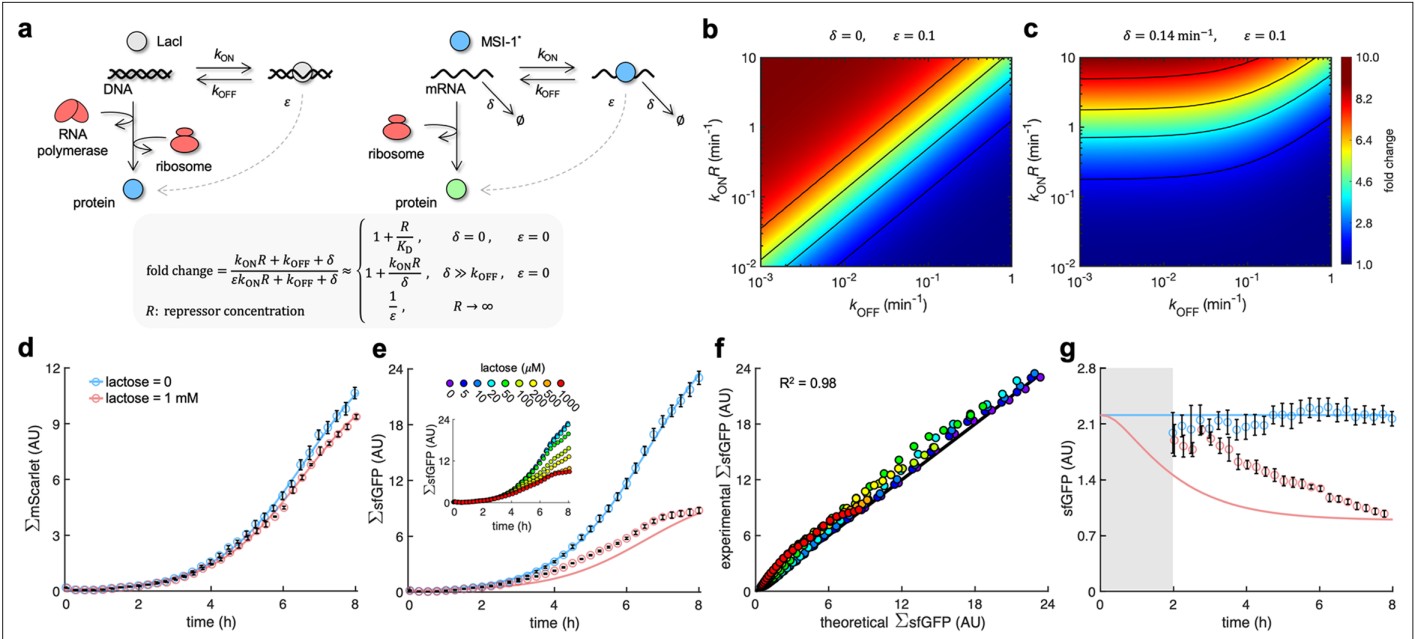

**Figure 3.** A mathematical model captures the dynamic response of the system. (**a**) Schematics of gene regulation at different levels with proteins that bind to nucleic acids (DNA or RNA). On the left, schematic of transcription regulation (e.g., LacI regulating MSI-1* expression). On the right, schematic of translation regulation (e.g., MSI-1* regulating sfGFP expression). A general mathematical expression (gray shaded) was derived to calculate the fold change in protein expression as a function of the regulator concentration ($R$), the association and dissociation rates ($k_{ON}$ and $k_{OFF}$), the elongation leakage fraction ($\varepsilon$), and the nucleic acid degradation rate ($\delta$). (**b**) Heatmap of the fold change as a function of $k_{ON}R$ and $k_{OFF}$ (i.e., the first-order kinetic rates that characterize the protein–DNA/RNA interaction) when $\delta = 0$ and $\varepsilon = 0.1$. This would correspond to transcription regulation. (**c**) Heatmap of the fold change when $\delta = 0.14$ min$^{-1}$ and $\varepsilon = 0.1$. This would correspond to translation regulation. (**d**) Total red fluorescence of the cell population ($\Sigma$ mScarlet) over time without and with 1 mM lactose. In this case, the cell growth rate was fitted to 0.80 h$^{-1}$. (**e**) Total green fluorescence of the cell population ($\Sigma$ sfGFP) over time without and with 1 mM lactose. The inset shows the dynamic response for different lactose concentrations. (**f**) Correlation between the experimental values of $\Sigma$ sfGFP at different times and for different lactose concentrations and the predicted values from a mathematical model that accounts for population growth and gene regulation. Data for $t > 2$ h. Linear regression performed. (**g**) Ratio of total green and red fluorescence as a proxy of cellular sfGFP expression over time. Ratio not represented at early times due to the high error obtained given the low number of cells present in the culture (gray shaded area). Deviations calculated by propagation. In all cases, points correspond to experimental data, while solid lines come from an adjusted mathematical model. Error bars correspond to standard deviations (n = 3). AU, arbitrary units.

The online version of this article includes the following source data and figure supplement(s) for figure 3:

**Source data 1.** Bulk fluorescence data of sfGFP and mScarlet with time.

**Figure supplement 1.** Characterization of the system response with IPTG (implemented with pRM1+ and pREP6).

**Figure supplement 2.** Dynamic response of the system in solid medium.

rate from the sequence (***Salis et al., 2009***). However, only a poor correlation ($R^2 = 0.16$) with the maximal expression level was observed (***Figure 2e***), suggesting that additional variables should be considered. For example, it was surprising the higher expression level in the case of mutant 4, despite a minimal change in the structure of the RNA motif (***Figure 2—figure supplement 3a***; we ensured that *sfGFP* was in frame in this case). On the other hand, when the fold change was correlated with the inverse of the dissociation constant ($1/K_D$, i.e., the equilibrium constant) better results were obtained ($R^2 = 0.75$; ***Figure 2f***). Mutant 1 is illustrative in this case because, even though a fast association rate was preserved (1.6 nM$^{-1}$ min$^{-1}$), it displayed a marginal regulatory activity as a result of a shorter residence time (0.41 min). This indicates that the underlying protein–RNA interaction in the bacterial circuit was close to thermodynamic equilibrium.

## A mathematical model captured the dynamic response of the system

Translation regulation is more challenging than transcription regulation because mRNA is unstable compared to DNA, especially in bacteria. In *E. coli*, in particular, the average mRNA half-life is about 5 min (***Bernstein et al., 2002***). However, it is possible to derive a common mathematical framework

from which to analyze the dynamics of both regulatory modes (*Figure 3a*). The fold change in protein expression is a suitable mesoscopic parameter that is directly related to the kinetic parameters that characterize the interaction in the cell (*Garcia and Phillips, 2011*). Using mass action kinetics, we obtained a general mathematical description of the fold change as a function of the regulator concentration ($R$), the association and dissociation rates, the leakage fraction of RNA/peptide-chain elongation, and the nucleic acid degradation rate (Appendix 2). To visualize the impact of the different parameters, we represented the fold change equation as a heatmap. When there is no nucleic acid degradation (DNA), a linear dependence between the first-order association rate ($k_{ON}R$) and $k_{OFF}$ is established to maintain a given fold change value (*Figure 3b*), which would correspond to the case of transcription regulation. Accordingly, our model converges to the classical description of fold = $1 + R/K_D$. However, if the nucleic acid degrades quickly (mRNA), the dependence between the first-order kinetic rates becomes nonlinear (*Figure 3c*). Indeed, in the case of translation regulation, it is important to note that when $k_{ON}R$ is lower than the mRNA degradation rate (i.e., the mRNA is degraded faster than the protein binds), the functionality is greatly compromised. To overcome this barrier, the regulator needs to be highly expressed as MSI-1* is in our system (we estimate $R > 1$ μM with 1 mM lactose). Furthermore, when the residence time is much longer than the mRNA half-life (i.e., the mRNA is degraded before the protein unbinds), $K_D$ is not a suitable parameter to characterize the regulation, which is solely association-dependent, resulting in non-equilibrium thermodynamics (*Goiriz and Rodrigo, 2021*). According to the aforementioned kinetic rates, this would be the case for MS2CP, but not for MSI-1* (i.e., both $k_{ON}$ and $k_{OFF}$ are instrumental to describe the regulation exerted by MSI-1*). Furthermore, given the 2.5-fold downregulation in our system, we estimated an elongation leakage fraction of 40% (using the fold change equation in the limit $R \to \infty$). This leakage would come from the ability of ribosomes to elongate even if MSI-1* is bound and their ability to bind sooner to the *sfGFP* mRNA due to a conserved transcription–translation coupling mechanism (*Kohler et al., 2017*).

In addition, we studied the transient response of the gene circuit with lactose as both MSI-1* and sfGFP expressions changed with time. For that, we quantified the total red fluorescence of the cell population (*Figure 3d*), which is an estimate of the total number of cells, and the total green fluorescence (*Figure 3e*), which comes from the composition of population growth and gene regulation. We developed a bottom-up mathematical model based on differential equations to predict sfGFP expression in the cell (Appendix 3), as well as a phenomenological model for the bacterial growth (Appendix 4). The parameter values were adjusted with the curves without and with 1 mM lactose. Then, we used the mathematical model to predict the transient responses for different intermediate lactose concentrations, finding excellent agreement with the experimental data ($R^2 = 0.98$; *Figure 3f*). We also characterized the time-course response of the circuit with IPTG, encountering similar results (*Figure 3—figure supplement 1*). Moreover, to explore the maintenance of the regulatory behavior when the cell physiology changes, we characterized cells growing in solid medium with a repurposed LigandTracer technology, which initially was developed to monitor molecular interactions in real time (*Björke and Andersson, 2006*). In this case, a significant difference in the total red fluorescence was observed without and with 1 mM IPTG, suggesting that MSI-1* expression was costly for the cell in these conditions. Besides, the total green fluorescence of the growing population was recapitulated using the model with a 2.6-fold downregulation of cellular sfGFP expression, which is in tune with the results in liquid medium (*Figure 3—figure supplement 2*). Subsequently, we analyzed the intracellular response. The time-dependent ratio of total green and red fluorescence was used as a proxy of sfGFP expression. A delay in the response is expected because MSI-1* needs to be produced upon addition of lactose (*Rosenfeld and Alon, 2003*). Nevertheless, our model predicted a faster response than experimentally observed (*Figure 3g*). Overall, this quantitative inspection of translation regulation backs connections between molecular attributes and cellular behavior.

## Rational redesign of the targeted transcript to enhance the dynamic range of the response

The presence of stem-loop structures in the N-terminal coding region contributes to lower the expression level. The more stable and closer to the start codon, the greater the impact on expression (*Paulus et al., 2004*). We hypothesized that, by destabilizing the RNA motif for MSI-1 binding, we would obtain an alternative regulatory system with higher expression levels. Accordingly, a new reporter

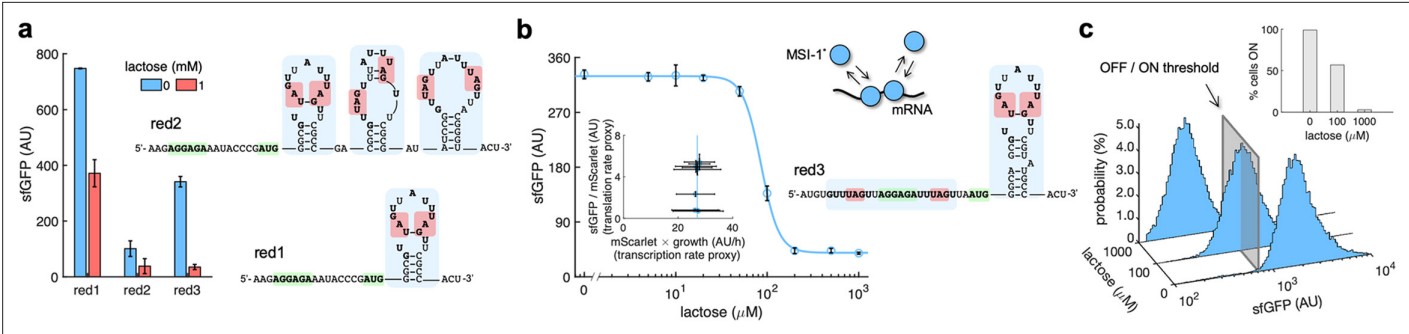

**Figure 4.** mRNA redesign to enhance the downregulation by Musashi-1. (**a**) Dynamic range of the response of three redesigned genetic systems using lactose (1 mM), showing a statistically significant regulation in the first and third cases (Welch's t-test, two-tailed p<0.05). The predicted secondary structures of the N-terminal coding regions of the reporter mRNAs are shown on the right. Redesign 1 (red1) was implemented with pREP4b and redesign 2 (red2) with pREP4b3x, which contains three MSI-1 binding sites. These stem-loop structures are less stable than the original one. Redesign 3 (red3) was implemented with pREP7. (**b**) Dose–response curve of the redesign-3 system using lactose as inducer (up to 1 mM). MSI-1* downregulated sfGFP expression by 8.6-fold (Welch's t-test, two-tailed p<0.05). The inset shows the scatter plot of the dynamic response in the Crick space (translation rate vs. transcription rate; vertical line fitted to 27 AU/h). The predicted secondary structure of the N-terminal coding region of the reporter mRNA is shown on the right; the mRNA contains two MSI-1 binding sites (blue shaded). In the 5′ UTR, the binding site is formed by two $RU_nAGU$ repeats that flank the RBS without forming secondary structure. In the N-terminal coding region, the binding site is the original one. The minimal cores (UAG) are marked in red. Points correspond to experimental data, while the solid line comes from an adjusted mathematical model. In all cases, error bars correspond to standard deviations (n = 3). (**c**) Probability-based histograms of sfGFP expression from single-cell data for different lactose concentrations (redesign 3), showing a statistically significant regulation (one-way ANOVA, p<10⁻⁴). The inset shows the percentage of cells in the ON state (sfGFP expressed), according to a specified threshold, for each lactose concentration. AU, arbitrary units.

The online version of this article includes the following source data and figure supplement(s) for figure 4:

**Source data 1.** Bulk fluorescence data of sfGFP with lactose.

**Figure supplement 1.** PLlac promoter tightly controls MSI-1* expression.

system was engineered removing three base pairs from the stem, maintaining the two consensus recognition sequences. An experimental analysis revealed a 4.9-fold increase of the maximal sfGFP expression level and a 2.0-fold downregulation with 1 mM lactose (*Figure 4a*, redesign 1). We then investigated the possibility of increasing the dynamic range of the response by placing three consecutive RNA motifs. However, we did not observe a greater downregulation with 1 mM lactose (*Figure 4a*, redesign 2), suggesting that the additional motifs far away from the start codon had no effect; what was noticed is an effect on the maximal expression level.

As a further strategy to enhance the dynamic range of the response, we redesigned the 5′ UTR of *sfGFP* to accommodate two additional $RU_nAGU$ repeats (viz., GUUUAGU and AUUUAGU) flanking the ribosome binding site (RBS), maintaining the original RNA motif after the start codon. In this way, MSI-1* can also block the RNA component of the 30S ribosomal subunit. Indeed, this is a widespread post-transcriptional regulatory strategy in prokaryotes, as it happens, for example, with the MS2 phage replicase (*Babitzke et al., 2009*). It is worth to note that the new 5′ UTR remained unstructured. We characterized by bulk fluorometry the dose-response curve of this new system, revealing an 8.6-fold downregulation of sfGFP expression by MSI-1* (*Figure 4b*, redesign 3; see also *Figure 4—figure supplement 1* to appreciate the tight control of MSI-1* expression with the PLlac promoter). This was a substantial increase in performance with respect to the 2.5-fold downregulation of the system shown in *Figure 1b*. Fitting a Hill equation, we obtained a regulatory coefficient of 86 μM and a Hill coefficient of 4.5 (Appendix 1). While the regulatory coefficient was similar than in the original system (99 μM), the Hill coefficient was significantly higher (compared to 1.7). Interestingly, an apparent cooperativity was established between two MSI-1* proteins by binding to adjacent sites. The dynamics of the system was also represented in the Crick space to highlight the change in translation rate. At the single-cell level, we found a 91% of ON cells in the uninduced state that decreased to 5.3% with 1 mM lactose (*Figure 4c*). Taken together, our data present MSI-1* as a powerful heterologous translation regulator in bacteria.

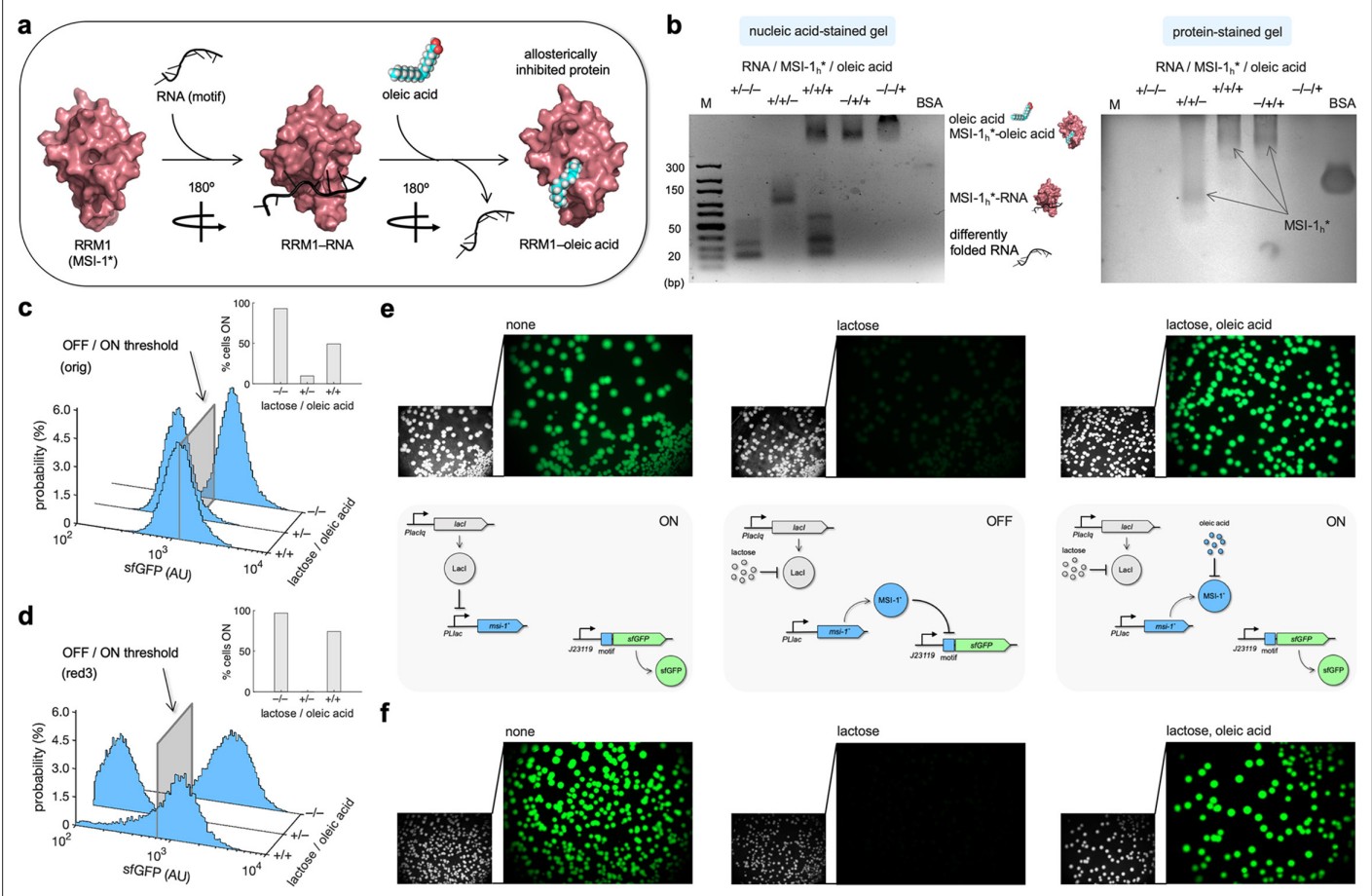

**Figure 5.** Oleic acid inhibits the regulatory activity of Musashi-1 in bacteria. (**a**) Three-dimensional structural schematic of the allosteric regulation. RRM1 of MSI-1 is shown alone, in complex with the RNA motif, and in complex with oleic acid. (**b**) Gel electrophoresis mobility shift assay to test the allosteric inhibition of MSI-1* with oleic acid. A purified MSI-1* protein (45 µM), the RNA motif as a label-free sRNA molecule (11 µM), and oleic acid (1 mM) were mixed in a combinatorial way in vitro. On the left, nucleic acid-stained gel. On the right, protein-stained gel (Coomassie). The different formed species are indicated. M denotes molecular marker (GeneRuler ultra-low range DNA ladder, 10–300 bp, Thermo). BSA was used as a control. (**c, d**) Probability-based histograms of sfGFP expression from single-cell data for different induction conditions (1 mM lactose or 1 mM lactose + 20 mM oleic acid) for the original system (**c**) and the redesign-3 system (**d**), showing statistically significant regulation in both cases (one-way ANOVA, $p < 10^{-4}$). The insets show the percentages of cells in the ON state (sfGFP expressed), according to a specified threshold, for each condition. (**e**) On the top, images of *E. coli* colonies harboring pRM1+ and pREP6. Bacteria were seeded in LB-agar plates with suitable inducers (1 mM lactose or 1 mM lactose + 20 mM oleic acid). Fluorescence and bright-field images are shown. On the bottom, schematics of the working modes of the synthetic gene circuit according to the different induction conditions. (**f**) Images of *E. coli* colonies harboring pRM1+ and pREP7. AU, arbitrary units.

The online version of this article includes the following source data and figure supplement(s) for figure 5:

**Source data 1.** Single-cell data of sfGFP.

**Source data 2.** Full gel images.

**Figure supplement 1.** Gel electrophoresis mobility shift assays to test the MSI-1$_h$*-RNA and the MSI-1$_h$*-oleic acid interactions (nucleic acid-stained gels).

**Figure supplement 1—source data 1.** Full gel images.

**Figure supplement 2.** 2D visualization of probability-based histograms of sfGFP expression from single-cell data.

**Figure supplement 3.** Quantification of the green fluorescence of the colonies (denoted by $\Sigma$ sfGFP as it is from populations; n = 5).

## The regulatory activity of a Musashi protein in bacteria can be externally controlled by a fatty acid

The ability of proteins to respond to small molecules is instrumental for environmental and metabolic sensing. Previous work revealed that MSI-1 can be allosterically inhibited by $\omega-9$ monounsaturated fatty acids and, in particular, by oleic acid (*Clingman et al., 2014*), an 18-carbon fatty acid

naturally found in various animal and plant oils (e.g., olive oil). Oleic acid binds to the RRM1 domain of MSI-1 and induces a conformational change that prevents RNA recognition (*Figure 5a*). To gain insight about the interactions between the elements of our system, we performed gel electrophoresis mobility shift assays using the purified MSI-1* protein, the RNA motif as a label-free sRNA molecule, and oleic acid. The different mobility of the nucleic acids upon binding to proteins and the coincident staining capacity of nucleic and fatty acids were exploited. We confirmed the MSI-1*–RNA interaction using a protein concentration gradient in this in vitro setup (*Figure 5—figure supplement 1a*), and we found that the interaction was completely disrupted in the presence of 1 mM oleic acid (*Figure 5b*). Furthermore, using an oleic acid concentration gradient, we obtained a half-maximal effective inhibitory concentration of about 0.5 mM (*Figure 5—figure supplement 1b*).

Subsequently, we assessed the effect of oleic acid over the regulatory activity of MSI-1* expressed in *E. coli*. This bacterium has evolved a machinery to uptake fatty acids from the environment. FadL and FadD are two membrane proteins that act as transporters, and FadE is the first enzyme that processes the fatty acid via the β-oxidation cycle (*Fujita et al., 2007*). Because of the high turbidity of the cell culture observed in the presence of oleic acid, we characterized the system by single-cell measurements of sfGFP expression by flow cytometry. In the case of the original system, the percentage of cells in the ON state increased from 10% (with 1 mM lactose) to 49% upon addition of 20 mM oleic acid (*Figure 5c*; see the 2D probability-based histograms in *Figure 5—figure supplement 2*). However, the initial 93% of ON cells observed in the absence of lactose was not recovered. Arguably, oleic acid was partially degraded once it entered the cell. Nevertheless, the system implemented with the redesign-3 reporter displayed a better dynamic behavior in response to lactose and oleic acid. In particular, the percentage of cells in the ON state increased from almost 0 (with 1 mM lactose) to 71% upon addition of 20 mM oleic acid (*Figure 5d*; see also *Figure 5—figure supplement 2*). In addition, we investigated this allosteric regulation by imaging the fluorescence of bacterial colonies grown in solid medium with different inducers. In stationary phase, FadE and the rest of oxidative

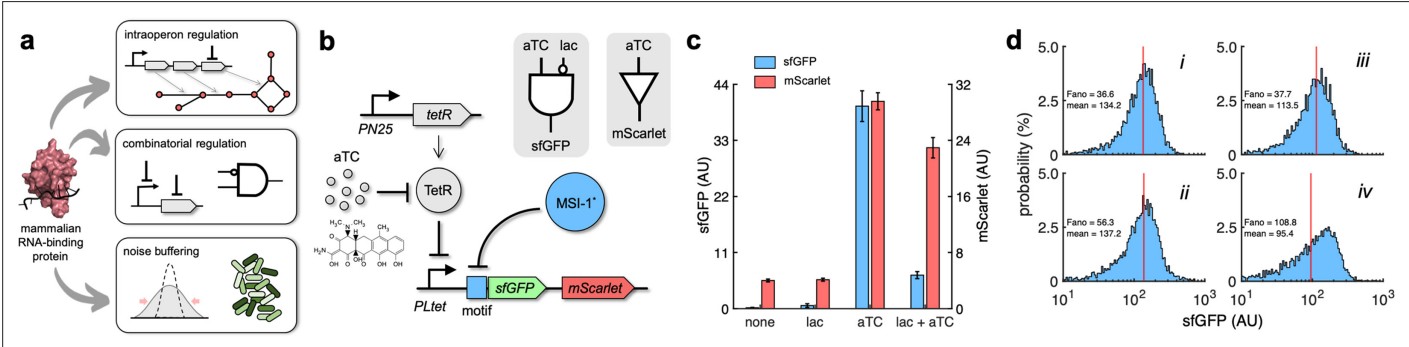

**Figure 6.** Applications of Musashi-1 for a fine expression control in bacteria. (**a**) Overview of the regulatory utility of MSI-1*. It could (i) regulate the expression of a given enzyme belonging to a polycistronic operon for a metabolic pathway control, (ii) be exploited together with transcription factors to implement combinatorial regulations following the genetic information flow, envisioning biosensing applications, and (iii) regulate noise in protein expression with the aim of producing cell populations less disperse, especially for bacterial delivery applications in animals. (**b**) Schematic of a new synthetic gene circuit engineered in *E. coli*. MSI-1* was always expressed from the PLlac promoter to be induced with lactose in a genetic background overexpressing LacI. sfGFP and mScarlet were used as reporters, both expressed from the PLtet promoter to be induced with anhydrotetracycline (aTC) in a genetic background overexpressing TetR. In this bicistronic operon, only sfGFP was under the control of a suitable RNA motif recognized by MSI-1* in the leader region of the transcript (original motif or redesign-3). In electronic terms, this circuit implements a NIMPLY logic gate considering sfGFP as the output. System implemented with pRM1+ and pREP6α or pREP7α. (**c**) Dynamic range of the response using lactose (1 mM) and aTC (100 ng/mL) in a combinatorial way. aTC significantly activated the expression of the operon (Welch's *t*-test, two-tailed p<0.05) and lactose, through the action of MSI-1*, significantly downregulated sfGFP expression in a specific way (data for pREP7α; Welch's *t*-test, two-tailed p<0.05). Error bars correspond to standard deviations (n = 3). (**d**) Probability-based histograms of sfGFP expression from single-cell data for different inducer concentrations. On the left, pRM1+ and pREP6α with 100 ng/mL aTC and 1 mM lactose (i) or 15 ng/mL aTC (ii). On the right, pRM1+ and pREP7α with 100 ng/mL aTC and 1 mM lactose (iii) or 30 ng/mL aTC (iv). The mean expression and Fano factor are shown. AU, arbitrary units.

The online version of this article includes the following source data and figure supplement(s) for figure 6:

**Source data 1.** Bulk fluorescence data of sfGFP and mScarlet and single-cell data of sfGFP.

**Figure supplement 1.** Probability-based histograms of sfGFP expression from single-cell data for different inducer concentrations (1 mM lactose + 20 mM oleic acid on the top, 0.1 mM lactose on the bottom).

enzymes could be saturated with the fatty acids generated from the membrane degradation (*Navarro Llorens et al., 2010*), oleic acid then having more time to interact with MSI-1*. Notably, we found a substantial inhibition of the repressive action of MSI-1* with 20 mM oleic acid in the case of both systems (*Figure 5e and f*; see also *Figure 5—figure supplement 3*). Conclusively, these results illustrate how the plasticity of RRM-containing proteins (e.g., MSI-1) can be exploited to engineer, even in simple organisms, gene regulatory circuits that operate in an integrated way at the transcriptional, translational, and post-translational levels.

## Application of a Musashi protein for intra-operon, combinatorial, and noise regulation

Transcription regulation has been engineered in *E. coli* to end with purposeful and versatile gene expression programs (*MacDonald et al., 2021*; *Nielsen et al., 2016*). However, this type of control faces limitations, such as to regulate a specific gene within an operon or to implement a definite combinatorial regulation without a large screening of promoter variants. To show that MSI-1* is instrumental to address these issues and ultimately increase our ability to program gene expression (*Figure 6a*), a new regulatory circuit was engineered in which *sfGFP* and *mScarlet* were both forming a single transcriptional unit (i.e., bicistronic operon) under a synthetic PL-based promoter regulated by the tetracycline repressor protein (TetR; promoter termed PLtet) (*Lutz and Bujard, 1997*). This allowed controlling the expression of both fluorescent proteins at the transcriptional level with anhydrotetracycline (aTC) in a genetic background overexpressing TetR. Furthermore, an RNA motif for MSI-1 binding was placed in front of *sfGFP* (*Figure 6b*). A characterization by bulk fluorometry using lactose (1 mM) and aTC (100 ng/mL) in a combinatorial way showed the specific regulation of sfGFP expression by MSI-1* and the ability to combine signals exploiting transcription and translation regulation (*Figure 6c*; implementation with the redesign-3 motif due to its enhanced dynamic range). A NIMPLY gate would model the logic behavior of the resulting circuit (i.e., sfGFP would only turn on with aTC and without lactose in the medium). These data also excluded the possibility that MSI-1* operated transcriptionally as a result of spurious DNA targeting.

In addition, we analyzed how MSI-1* regulated noise in protein expression monitoring green fluorescence in single cells. Inducing the circuit of *Figure 6b* with 100 ng/mL aTC and 1 mM lactose produced almost the same mean expression level than with an intermediate aTC concentration (15 ng/mL when the implementation was with the original motif and 30 ng/mL when it was with the redesign-3 motif). However, the resulting unimodal distributions displayed different dispersions, lower when MSI-1* was not repressed. The Fano factor (the ratio between variance and mean) (*Sanchez et al., 2013*) was used to quantify the responses, finding reductions of 35 and 65% depending on the implementation (*Figure 6d*). Furthermore, for the circuit of *Figure 1b*, we found a 38% lower Fano factor when inducing with 1 mM lactose and 20 mM oleic acid than with 0.1 mM lactose, despite having similar mean expression levels (*Figure 6—figure supplement 1*). Of note, the response sensitivity was dominated by transcription regulation when the PL-based promoter was induced with an intermediate concentration of lactose (0.1 mM) or aTC (15–30 ng/mL). By contrast, the response sensitivity was dominated by translation regulation when the PL-based promoter was fully induced (1 mM lactose or 100 ng/mL aTC), thereby controlling the heterogeneity of the response (*Dolcemascolo et al., 2022*). Overall, these results illustrate the utility of repurposed mammalian RNA-binding proteins in bacteria for a fine expression control.

## Discussion

The successful incorporation of the mammalian MSI-1 protein as a translation factor in *E. coli* highlights, in first place, the versatility of RRM-containing proteins to function as specific post-transcriptional regulators in any living cell, from prokaryotes to eukaryotes. Our data show that the protein–RNA association phase is very fast, which is suitable for regulation even in cellular contexts in which RNA molecules are short-lived, such as in *E. coli* (*Bernstein et al., 2002*). Nonetheless, it is important to stress that the kinetic parameters in vivo might differ from those measured in vitro due to off-target bindings and crowding effects (*Hammar et al., 2012*). Moreover, our data show that a downregulation of translation rate up to 8.6-fold can be achieved, with an appropriate design of the target mRNA leader region, and that the engineered cell can sense oleic acid from the environment. Here, the

C-terminal low-complexity domain of the native MSI-1 was discarded to create MSI-1* (*Iwaoka et al., 2017*), in order to increase solubility, even though this domain might contribute to RNA binding (*Järvelin et al., 2016*). Further work should be conducted to enhance the fold change of the regulatory module and engineer complex circuits with it.

Interestingly, proteins associated with clustered regularly interspaced short palindromic repeats (CRISPR), which belong to the prokaryotic immune system, contain distorted RRM versions (*Koonin and Makarova, 2013*). Some CRISPR proteins might have evolved, for example, from an ancestral RRM-based (palm) polymerase after duplications, fusions, and diversification. Noting that the palm domain indeed presents an RRM-like fold (*Anantharaman et al., 2010*), we hypothesize that a boost of functionally diverse RRM-containing proteins took place once the polymerases were confined into the nucleus, as the pressure for efficient replication was relieved in the cytoplasm, which would provide a rationale on the unbalance noticed between eukaryotes and prokaryotes (*Maris et al., 2005*; *Koonin et al., 2020*).

In second place, our results pave the way for engineering more complex circuits in bacteria with plastic and orthogonal RNA-binding proteins, such as MSI-1, capable of signal multiplexing. Nature is a formidable reservoir of functional genetic material sculpted by evolution that can be exploited to (re)program specific living cells (*Khalil and Collins, 2010*). However, to overcome biological barriers, transgenes usually come from related organisms or cognate parasites at the cost of limiting the potential engineering. Therefore, efforts to borrow functional elements from highly diverse organisms are suggestive (e.g., regulatory proteins from mammals to bacteria), with the ultimate goal of developing industrial or biomedical applications.

Notably, advances in synthetic biology have pushed the bioproduction of a wide variety of compounds in bacteria as a result of a better ability to fine-tune enzyme expression (*Choi et al., 2019*). Translation regulation is instrumental to this end because in multiple cases different enzymes are expressed from the same transcriptional unit (i.e., operon). Previous work exploited regulatory RNAs for such a tuning (*Na et al., 2013*), but the use of RNA-binding proteins as translation factors is also appealing. We envision the application of MSI-1* as a genetic tool for metabolic engineering. The additional use of RNA-binding proteins able to alter mRNA stability might lead to the implementation of more complex circuits at the post-transcriptional level. Furthermore, MSI-1* is able to respond to fatty acids, which are ideal precursors of potential biofuels due to their long hydrocarbon chains. In particular, biofuel in the form of fatty acid ethyl ester, whose bioproduction in *E. coli* can be optimized by reengineering the regulation of the β-oxidation cycle with the allosteric transcription factor FadR (*Zhang et al., 2012*). Arguably, MSI-1* might be used in place of or in combination with FadR for subsequent developments. However, engineering regulatory circuits for efficient bioproduction is not evident in general as the enzymatic expression levels may require fine-tuning, so systems-level mathematical models need to be considered for design along with a wide genetic toolkit for implementation (*Choi et al., 2019*). We anticipate that other animal RRM-containing proteins might be repurposed in *E. coli* as translation factors. Moreover, protein design might be used to reengineer MSI-1* in order to respond to new ligands, maintaining high specificity and affinity for a particular RNA sequence, as previously done with the transcription factor LacI (*Taylor et al., 2016*).

In addition, the Musashi protein family is of clinical importance, as in humans it is involved in different neurodegenerative disorders (e.g., Alzheimer's disease) and some types of cancer (*Fox et al., 2015*; *Montalbano et al., 2020*; *Kang et al., 2017*). Therefore, the development of simple genetic systems from which to test protein mutants, potential target mRNAs, decoying RNA aptamers, and inhibitory small molecules in a systematic manner is very relevant. Furthermore, isolating human regulatory elements would help to filter out indirect effects that likely occur in the natural context. This might lead to new therapeutic opportunities. Nevertheless, one limitation of using *E. coli* as a chassis is that some post-translational modifications (PTMs) may be lost, thereby compromising the functionality of the expressed proteins (*Sahdev et al., 2008*). Fortunately, there are metabolic engineering efforts devoted to implement eukaryotic PTM pathways in *E. coli*, such as the glycosylation pathway (*Valderrama-Rincon et al., 2012*).

In conclusion, the functionalization of RRM-containing proteins in bacteria offers exciting prospects, especially as more information becomes available on how individual RRM domains bind to precise RNA sequences, interact with further protein domains, and respond to small molecules through allosteric effects. This work illustrates how synthetic biology, through the rational assembly

of heterologous genes and designer *cis*-regulatory elements into circuits, is useful to generate knowledge about the application range of a fundamental type of proteins in nature.

## Materials and methods
### Strains, plasmids, and reagents

*E. coli* Dh5α was used for cloning purposes following standard procedures. To express our genetic circuit for functional characterization, *E. coli* MG1655-Z1 cells (*lacI*[+], *tetR*[+]) were used. This strain was co-transformed with two plasmids, called pRM1+ (KanR, pSC101-E93R ori; leading to ~230 copies/cell) (*Peterson and Phillips, 2008*) and pREP6 (CamR, p15A ori; leading to ~15 copies/cell). On the one hand, pRM1+ was obtained by cloning a truncated coding region of the *M. musculus* MSI-1 protein (the first 192 amino acids, containing the two RRMs; UniProt #Q61474; termed MSI-1*). This gene was under the transcriptional control of the inducible promoter PLlac. On the other hand, pREP6 was obtained by cloning the coding region of sfGFP with an RNA sequence motif recognized by MSI-1. The coding region of mScarlet was also present in the plasmid. These two genes were under the control of the constitutive promoter J23119 in two different transcriptional units. In addition to the original RNA sequence motif, five point-mutated sequences were designed and cloned in pREP6. Additional RNA sequence motifs were cloned in front of *sfGFP* for control experiments (the resulting plasmids were named pREP4, pREP4b, pREP4b3x, and pREP7). In particular, pREP4b3x incorporates three RNA motifs in tandem after the start codon, and pREP7 has two $RU_nAGU$ repeats flanking the RBS and a full RNA motif after the start codon. Additional reporter plasmids were constructed using the inducible promoter PLtet to assess the intra-operon regulation, the implementation of combinatorial regulation, and the buffering of expression noise (the resulting plasmids were named pREP6α and pREP7α). Suitable genetic cassettes to obtain the final constructions were synthesized by IDT. Appendix 5 lists all plasmids used in this work. Appendix 6 presents the nucleotide sequences of the different genetic elements.

To perform the dynamic assays with LigandTracer (Ridgeview), *E. coli* BL21(DE3) cells (*lacI*[+], *T7pol*[+]) were used. This strain was also co-transformed with pRM1+ and pREP6. To purify a recombinant Musashi protein, *E. coli* BL21-Gold(DE3) cells (*lacI*[+], *T7pol*[+]) were used. A truncated coding region of the human MSI-1 protein (the first 200 amino acids; UniProt #O43347; termed MSI-1$_h$*) was cloned under the control of a T7pol promoter into the plasmid pET29b (KanR, pUC ori).

Luria-Bertani (LB) medium was used for the overnight cultures and M9 minimal medium (1× M9 minimal salts, 2 mM $MgSO_4$, 0.1 mM $CaCl_2$, 0.05% thiamine, 0.05% casamino acids, and 1% glycerol or 0.4% glucose) for the characterization cultures. M9-glucose medium was only used for real-time fluorescence quantification in liquid medium with IPTG. LB-agar was used for real-time fluorescence quantification in solid medium. Kanamycin and chloramphenicol were used at a concentration of 50 µg/mL and 34 µg/mL, respectively. Lactose and IPTG were used as the inducers of the system (controlling the expression of MSI-1* in *E. coli*) at a concentration of 5, 10, 20, 50, 100, 200, 500, or 1000 µM. aTC was also used to induce the modified systems with PLtet at a concentration of 15, 30, or 100 ng/mL. Oleic acid was used as the allosteric inhibitor of MSI-1* at a concentration of 20 mM in the in vivo assays (both in liquid and solid medium). In the in vitro assays, oleic acid was used at a concentration of 0.01, 0.1, 0.2, 0.5, 0.7, 1, 1.5, or 2 mM. It was neutralized with NaOH and used in a medium containing 0.5% tergitol NP-40. Compounds were provided by Merck.

### Bulk fluorometry

Cultures (2 mL) inoculated from single colonies (three replicates) were grown overnight in LB medium with shaking (220 rpm) at 37 °C. Cultures were then diluted 1:100 in fresh M9 medium (200 µL) with the appropriate inducer (lactose, IPTG, and/or aTC). The microplate (96 wells, black, clear bottom; Corning) was incubated with shaking (1300 rpm) at 37 °C up to 8–10 h (to reach an $OD_{600}$ around 0.5–0.7). At different times, the microplate was assayed in a Varioskan Lux fluorometer (Thermo) to measure absorbance (600 nm), green fluorescence (excitation: 485 nm, emission: 535 nm), and red fluorescence (excitation: 570 nm, emission: 610 nm). To characterize the time-course response of the system, cultures were grown to exponential phase and then diluted before adding the inducer (to minimize the response lag). Mean background values of absorbance and fluorescence, corresponding to M9 medium, were subtracted to correct the signals. Normalized fluorescence was calculated as the

slope of the linear regression between fluorescence and absorbance (assuming fluorophore maturation faster than cell doubling time and no proteolytic degradation) (*Leveau and Lindow, 2001*). The mean value of normalized fluorescence corresponding to non-transformed cells was then subtracted to obtain a final estimate of expression. In addition, cell growth rate was calculated as the slope of the linear regression between the logarithm of background-subtracted absorbance and time in the exponential phase.

## Real-time fluorescence quantification in solid medium

Cultures (2 mL) inoculated from single colonies (three replicates) were grown overnight in LB medium with shaking (220 rpm) at 37 °C. The overnight culture was plated (15 µL) in areas A and D of a Multi-Dish 2x2 plate (Ridgeview) coated with LB-agar. IPTG was added in areas A and B of the dish at the final concentration of 1 mM. Area C was kept free of cells/inducers as a reference. The dish was then placed in the rotating support of the LigandTracer instrument (Ridgeview) and incubated at 37 °C for 24 h. The fluorescence from sfGFP and mScarlet was quantified with time in the seeded areas of the dish using the BlueGreen (excitation: 488 nm, emission: 535 nm) and OrangeRed (excitation: 568 nm, emission: 620 nm) detectors. The readouts of the opposite parts of the dish were subtracted to correct the signals.

## Flow cytometry

Cultures (2 mL) inoculated from single colonies (three replicates) were grown overnight in LB medium with shaking (220 rpm) at 37 °C. Cultures were then diluted 1:100 in fresh LB medium (200 µL) to load a microplate (96 wells, black, clear bottom; Corning) with the appropriate concentrations of lactose (0, 100, 1000 µM), oleic acid (0, 20 mM), and/or aTC (15, 30, 100 ng/mL). The microplate was then incubated with shaking (1300 rpm) at 37 °C until cultures reached a sufficient $OD_{600}$. Cultures (6 µL) were then diluted in PBS (1 mL). Fluorescence was measured in an LSRFortessa flow cytometer (BD) using a 488 nm laser and a 530 nm filter for green fluorescence. Events were gated by using the forward and side scatter signals and compensated (~$10^4$ events after this process). The mean value of the autofluorescence of the cells was subtracted to obtain a final estimate of expression. Data analysis was performed with MATLAB (MathWorks).

## Purification of a Musashi protein

Cells were grown in LB medium with shaking at 37 °C until $OD_{600}$ reached 0.6–0.8. Subsequently, the expression of MSI-1$_h$* was induced with 0.5 mM IPTG. Cells were incubated at 37 °C for 4 h and harvested by centrifugation at 7500 rpm for 15 min at 4 °C. The cell pellet was resuspended in a lysis buffer (50 mM Tris–HCl, pH 8.0, 500 mM NaCl, 10% glycerol, with protease inhibitor cocktail), ruptured by sonication, and separated by centrifugation at 30,000 rpm for 35 min at 4 °C. The soluble fraction was collected and treated with a 5% polyethylenimine solution in order to remove DNA/RNA attached to the protein. Resuspension of the protein was done in 20 mM Tris–HCl, pH 9.0, with protease inhibitor cocktail. Soluble protein was filtered with a 0.22 µm membrane and purified by ion-exchange chromatography using an Anion exchange Q FF 16/10 column previously equilibrated in alkaline buffer. The protein was collected on the flow-through. The protein was filtered and further purified to homogeneity by size-exclusion chromatography using a Hi load 26/60 Superdex 75 pg column previously equilibrated in alkaline buffer with NaCl. The purified fractions were collected and buffer exchange chromatography was performed using a HiPrep 26/10 Desalting column previously equilibrated with the final buffer (20 mM MES, pH 6.0, 100 mM NaCl, 0.5 mM EDTA, with protease inhibitor cocktail). Purification performed at Giotto.

## Binding kinetics assays of protein–RNA interactions

Binding experiments of the purified MSI-1$_h$* protein against different RNA ligands were performed using the switchSENSE proximity sensing technology (*Cléry et al., 2017*; *Langer et al., 2013*) and a suitable adapter chip on the heliX biosensor platform (Dynamic Biosensors). The adapter chip consists of a microfluidic channel with two gold electrodes functionalized with fluorophore-decorated DNA nanolevers that serve as linkers between the gold surface and the ligand of interest. A constant negative voltage is applied to the electrodes to keep the DNA nanolevers in an upright position. Binding between the injected analyte (MSI-1$_h$*) and the ligand attached to the sensor surface (RNA) leads to

the alteration of the chemical surrounding of the dye, which results in a fluorescence change. Fluorescence change of the dye in real time describes the binding kinetics of the molecule of interest. Kinetic experiments consisted of a protein association phase (5 min) and a dissociation phase (15 min) in which the chip was rinsed with a buffer (50 mM Tris–HCl, 0.5 mM EDTA, 140 mM NaCl, 0.05% Tween 20, 1 mM TCEP, pH 7.2). A flow rate of 100 µL/min was applied and a sampling rate of 1 Hz was used.

Six different RNA ligands (original and five mutants) were attached to the 5′ end of a generic 48 nt DNA ligand strand, which is part of the DNA linker system on the heliX adapter chip surface. All oligonucleotides were synthesized by Ella Biotech. The ligand strand was hybridized with an adapter strand carrying the fluorophore. Different fluorophores were tested toward their sensitivity for protein–RNA interactions. The green fluorophore Gb showed the most significant signal change. The other half of the adapter strand is complementary to a DNA anchor strand, which is pre-attached to the chip surface. The immobilization of the RNA used a standard functionalization procedure on the heliX device. Kinetic rate constants and affinities were obtained by fitting the experimental data with theoretical binding models implemented in the heliOS software (Dynamic Biosensors). Exponential decay models were used. As a negative control to check for unspecific protein–RNA binding, the single-strand RNA sequence CGGCGCCGC was used (without any binding motif). All data were referenced with a blank run and with the negative control.

## RT-qPCR

Cultures (2 mL) inoculated from single colonies (three replicates) were grown overnight in LB medium with shaking (220 rpm) at 37 °C. Cultures were then diluted 1:100 in fresh LB medium (2 mL) with the appropriate inducer (lactose) and were grown until $OD_{600}$ reached 0.6–0.8. Then, 500 µL of each culture was mixed with RNAprotect Bacteria Reagent (QIAGEN). Subsequently, RNA extraction was carried out with the RNeasy kit (QIAGEN), choosing the enzymatic lysis and proteinase K digestion of bacteria (recommended for Gram-negative bacteria grown in complex media). The eluted RNA sampled were quantified using a NanoDrop spectrophotometer (Thermo).

The TaqPath 1-step RT-qPCR master mix, CG was used. Then, 1 µL of sample was mixed with 500 nM of forward and reverse primers, 250 nM of ssDNA probe, and 5 µL of the master mix for a total volume of 20 µL (adjusted with RNase-free water) in a fast microplate (Applied). Two independent mixes were prepared, one for targeting *sfGFP* and another for the *E. coli* b3500 gene, which was employed as the reference gene. Reactions were performed in a QuantStudio 3 equipment (Thermo) with this protocol: incubation at 25 °C for 2 min for uracil-N glycosylation, followed by 50 °C for 15 min for RT (reverse transcription), followed by an inactivation step at 95 °C for 2 min, then followed by 40 cycles of amplification at 95 °C for 3 s and 60 °C for 30 s.

## Gel electrophoresis

Mobility shift assays with the purified MSI-1$_h$* protein and its cognate RNA motif were performed. The RNA motif was generated by in vitro transcription with the TranscriptAid T7 high yield transcription kit (Thermo) from a DNA template. It was then purified using the RNA clean and concentrator column (Zymo) and quantified in a NanoDrop spectrophotometer (Thermo). Bovine serum albumin (BSA) was used as a control protein (at 30 µM). Reactions with different combinations of elements were prepared (MSI-1$_h$* at 45 µM, RNA at 11 µM, and oleic acid at 1 mM). Reactions with concentration gradients of MSI-1$_h$* (from 0 to 45 µM) and oleic acid (from 0 to 2 mM) were also performed. Reactions were incubated for 30 min at 37 °C. Reaction volumes were then loaded in 3% agarose gels prepared with 0.5× TBE and stained using RealSafe (Durviz). Gels ran for 45 min at room temperature applying 110 V. The GeneRuler ultra-low-range DNA ladder (10–300 bp, Thermo) was used. This staining served to reveal the RNA and oleic acid (free or in complex with the MSI-1$_h$* protein) (**Perea and Greenbaum, 2020**; **Fessenden-Raden, 1972**). In addition, gels were soaked for 10 min in the Coomassie blue stain (Fisher) at room temperature with shaking to reveal the proteins. Gels were then soaked in a destaining solution overnight to remove the excess of blue stain. Pictures were taken with the Imager2 gel documentation system (VWR).

## Microscopy

LB-agar plates seeded with *E. coli* MG1655-Z1 cells co-transformed with pRM1+ and pREP6 or pREP7 were grown overnight at 37 °C. Lactose (1 mM) and oleic acid (20 mM) were used as supplements.

The plates were irradiated with blue light and images were acquired with a 2.8 Mpixel camera with a filter for green fluorescence in a light microscope (Leica MSV269). The commercial software provided by Leica was used to adjust the visualization of the differential fluorescence among plates. The fluorescence intensity of the colonies was quantified with Fiji (*Schindelin et al., 2012*).

## Mathematical modeling

On the one hand, Hill equations were used to empirically model sfGFP expression with lactose/IPTG, eBFP2 expression with lactose, and sfGFP expression with eBFP2 expression (see Appendix 1 for details). On the other hand, a system of ordinary differential equations was developed to model the dynamic response of the synthetic gene circuit from a bottom-up approach. The system accounted for the intracellular mRNA and protein concentrations, considering a scenario of equilibrium to model both LacI-DNA and MSI-1*-RNA binding (see Appendix 3 for details). Parameter values were obtained by nonlinear fitting against our experimental data.

## Molecular visualization in silico

The RMM1 of MSI-1 protein structure determined by nuclear magnetic resonance was downloaded from the UniProt database (https://www.uniprot.org/; *Bairoch et al., 2005*). A 3D structure of the RNA motif subsequence involving the two $RU_nAGU$ repeats was predicted with the RNAComposer software (*Popenda et al., 2012*). The oleic acid molecule was downloaded from the ChemSpider database (https://www.chemspider.com/). All the molecules were loaded, visualized, colored, trimmed (where necessary), and manually docked using the open-source PyMol software (Schrödinger; pymol.org).

## Resources availability

The sequences of all genetic elements used in this work are presented in the Appendixes. Plasmids available upon request to the corresponding author.

## Acknowledgements

We thank M Sattler (TUM) for useful discussions. This work was supported by the grants H2020-MSCA-ITN-2018 #813239 (RNAct) from the European Commission and PGC2018-101410-B-I00 (SYSY-RNA) from the Spanish Ministry of Science and Innovation (co-financed by the European Regional Development Fund). RD, APR, RAHR, and GPR acknowledge each a Marie Curie fellowship. LG was supported by a predoctoral fellowship from the Valencia Regional Government (ACIF/2021/183).

## Additional information

### Competing interests

Anna Pérez-Ràfols: APR works for Giotto Biotech. R Anahí Higuera-Rodríguez: RAHR works for Dynamic Biosensors. Guillermo Pérez-Ropero: GPR works for Ridgeview Instruments. Tommaso Martelli: TM works for Giotto Biotech. Wolfgang Kaiser: WFV works for Dynamic Biosensors. Jos Buijs: JB works for Ridgeview Instruments. The other authors declare that no competing interests exist.

### Funding

| Funder | Grant reference number | Author |
|---|---|---|
| European Commission | H2020-MSCA-ITN-2018 #813239 | Wim F Vranken Tommaso Martelli Wolfgang Kaiser Jos Buijs Guillermo Rodrigo |
| Ministerio de Ciencia e Innovación | PGC2018-101410-B-I00 | Guillermo Rodrigo |
| Generalitat Valenciana | ACIF/2021/183 | Lucas Goiriz |

| Funder | Grant reference number | Author |
|--------|------------------------|--------|

The funders had no role in study design, data collection and interpretation, or the decision to submit the work for publication.

## Author contributions

Roswitha Dolcemascolo, María Heras-Hernández, Lucas Goiriz, Formal analysis, Investigation, Writing – review and editing; Roser Montagud-Martínez, Supervision, Investigation; Alejandro Requena-Menéndez, Anna Pérez-Ràfols, Investigation; Raúl Ruiz, R Anahí Higuera-Rodríguez, Guillermo Pérez-Ropero, Formal analysis, Investigation; Wim F Vranken, Formal analysis, Supervision, Funding acquisition, Writing – review and editing; Tommaso Martelli, Wolfgang Kaiser, Jos Buijs, Formal analysis, Supervision, Funding acquisition; Guillermo Rodrigo, Conceptualization, Formal analysis, Supervision, Funding acquisition, Investigation, Writing - original draft

## Author ORCIDs

Wim F Vranken ![ORCID] https://orcid.org/0000-0001-7470-4324
Guillermo Rodrigo ![ORCID] http://orcid.org/0000-0002-1871-9617

Joint Public Review: https://doi.org/10.7554/eLife.91777.3.sa1
Author Response https://doi.org/10.7554/eLife.91777.3.sa2

# Additional files

## Supplementary files

• MDAR checklist

## Data availability

All data generated or analysed during this study are included in the manuscript and supporting files.

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

## Appendix 1

The repression of sfGFP as a function of lactose is modeled by the following Hill equation (**Weiss, 1997**)

$$[\text{sfGFP}] = \frac{A_1}{1 + \left(\dfrac{[\text{Lactose}]}{K_1}\right)^{n_1}} + B_1 \,,$$

where $K_1$ is the regulatory coefficient, $n_1$ the Hill coefficient, $A_1 + B_1$ the maximal expression level, and $B_1$ the basal expression level at full repression. In the case of sfGFP, its concentration is given by the normalized green fluorescence signal in arbitrary units (AU). The adjusted parameter values are $A_1 = 90.7$ AU, $B_1 = 62.1$ AU, $K_1 = 99.1$ M, and $n_1 = 1.70$ (**Figure 1c**).

We used the very same equation to model the dynamic response of the system implemented with pREP7. In this case, the adjusted parameter values are $A_1 = 289.9$ AU, $B_1 = 39.4$ AU, $K_1 = 85.7$ M, and $n_1 = 4.45$ (**Figure 4b**).

Also, the following Hill equation models the repression of sfGFP by IPTG

$$[\text{sfGFP}] = \frac{A_2}{1 + \left(\dfrac{[\text{IPTG}]}{K_2}\right)^{n_2}} + B_2 \,.$$

The adjusted parameter values are $A_2 = 76.4$ AU, $B_2 = 52.3$ AU, $K_2 = 71.3$ M, and $n_2 = 2.28$ (**Figure 3—figure supplement 1a**).

In addition, the activation of eBFP2, proxy of MSI-1*, as a function of lactose is modeled by the following Hill equation:

$$[\text{MSI-1*}] \propto [\text{eBFP2}] = \frac{A_3 \left(\dfrac{[\text{Lactose}]}{K_3}\right)^{n_3}}{1 + \left(\dfrac{[\text{Lactose}]}{K_3}\right)^{n_3}} + B_3 \,,$$

where $K_3$ is the regulatory coefficient, $n_3$ the Hill coefficient, $A_3 + B_3$ the maximal expression level, and $B_3$ the basal expression level with no activation. In the case of eBFP2, its concentration is given by the normalized blue fluorescence signal in AU. The adjusted parameter values are $A_3 = 23.1$ AU, $B_3 = 1.88$ AU, $K_3 = 359$ M, and $n_3 = 2.81$ (**Figure 1d**).

Finally, the following Michaelis equation (a particular case of the Hill equation when there is no cooperativity)

$$[\text{sfGFP}] = \frac{A_4}{1 + \dfrac{[\text{eBFP2}]}{K_4}}$$

defines the engineered regulation between MSI-1* (given by eBFP2) and sfGFP. Here, no basal expression level is considered. The adjusted parameter values are $A_4 = 165$ AU and $K_4 = 10.2$ AU (**Figure 1d**).

## Appendix 2

The fold change in protein expression can be calculated from the fundamental parameters that model the regulatory system, such as the association rate of the regulator to the nucleic acid ($k_{ON}$), the dissociation rate ($k_{OFF}$), the concentration of the regulator in the cell ($R$), and the degradation rate of the nucleic acid ($\delta$). If we denote by $A_0$ the concentration of free nucleic acid, by $A_R$ the concentration of nucleic acid with the regulator bound, and by $P$ the concentration of the regulated protein, we can write

$$\frac{dA_0}{dt} = \alpha - k_{ON}RA_0 + k_{OFF}A_R - \delta A_0$$

$$\frac{dA_R}{dt} = k_{ON}RA_0 - k_{OFF}A_R - \delta A_R$$

$$\frac{dP}{dt} = \beta A_0 + \varepsilon \beta A_R - \mu P,$$

where $\alpha$ is the synthesis rate of the nucleic acid, $\beta$ the synthesis rate of the protein, and $\varepsilon$ the leakage fraction of protein synthesis when the regulator is bound. Note that in steady state $A_{0\infty} + A_{R\infty} = \frac{\alpha}{\delta}$.

If $R = 0$, then $P_\infty = \frac{\alpha\beta}{\delta\mu}$ (steady state). If $R > 0$, then $P_\infty = \frac{\alpha\beta}{\delta\mu}\left(\frac{\varepsilon k_{ON}R + k_{OFF} + \delta}{k_{ON}R + k_{OFF} + \delta}\right)$. Therefore, it turns out that

$$\text{fold} = \frac{P_\infty(R = 0)}{P_\infty(R > 0)} = \frac{k_{ON}R + k_{OFF} + \delta}{\varepsilon k_{ON}R + k_{OFF} + \delta}.$$

Importantly, this model can be applied either to transcription regulation or translation regulation. The main difference is that in the case of transcription the nucleic acid targeted by the regulator (DNA) is stable (we can model this as $\delta = \mu$, and then set $\delta \simeq 0$ in the fold change equation), while in the case of translation, the nucleic acid targeted by the regulator (mRNA) is unstable ($\delta \gg \mu$).

## Appendix 3

The system of ordinary differential equations (ODEs) that governs the dynamics of the engineered circuit, considering the intracellular concentrations of mRNAs and proteins (*Rodrigo et al., 2011*), reads

$$\frac{d[\text{mRNA}_{\text{MSI-1*}}]}{dt} = \alpha_x \left( \frac{\rho_x + \left( \frac{[\text{Lactose}]}{\theta_x} \right)^{n_x}}{1 + \left( \frac{[\text{Lactose}]}{\theta_x} \right)^{n_x}} \right) - \delta[\text{mRNA}_{\text{MSI-1*}}]$$

$$\frac{d[\text{MSI-1*}]}{dt} = \beta_x[\text{mRNA}_{\text{MSI-1*}}] - \mu[\text{MSI-1*}]$$

$$\frac{d[\text{mRNA}_{\text{sfGFP}}]}{dt} = \alpha_y - \delta[\text{mRNA}_{\text{sfGFP}}]$$

$$\frac{d[\text{sfGFP}]}{dt} = \beta_y \left( \frac{1}{1 + \frac{[\text{MSI-1*}]}{\theta_y}} \right) [\text{mRNA}_{\text{sfGFP}}] - \mu[\text{sfGFP}],$$

where $\alpha_x$ is the maximal transcription rate of the *msi-1\** gene, $\alpha_y$ the maximal transcription rate of the *sfGFP* gene, $\beta_x$ the maximal translation rate of *msi-1\**, $\beta_y$ the maximal translation rate of *sfGFP*, $\delta$ the mRNA degradation rate (assumed equal for the *msi-1\** and *sfGFP* genes), $\rho_x$ the repression fold of LacI at the transcriptional level, $\theta_x$ the effective dissociation constant between LacI and lactose, $n_x$ the effective binding cooperativity of LacI, $\theta_y$ the effective dissociation constant between MSI-1\* and the RNA motif in the *sfGFP* gene, and μ the bacterial growth rate.

The analytical solution of this system of ODEs can be obtained through the use of the Laplace transform (*Bracewell, 2000*) and reads

$$[\text{mRNA}_{\text{MSI-1*}}](t) = \frac{\alpha_x}{\delta} \left( \frac{\rho_x + \left( \frac{[\text{Lactose}]}{\theta_x} \right)^{n_x}}{1 + \left( \frac{[\text{Lactose}]}{\theta_x} \right)^{n_x}} \right) \left( 1 - e^{-\delta t} \right) + [\text{mRNA}_{\text{MSI-1*}}]_0 e^{-\delta t}$$

$$[\text{MSI-1*}](t) = \beta_x \int_0^t e^{-\mu(t-\tau)} [\text{mRNA}_{\text{MSI-1*}}](\tau) d\tau + [\text{MSI-1*}]_0 e^{-\mu t} \simeq$$
$$\simeq \frac{\beta_x}{\mu} [\text{mRNA}_{\text{MSI-1*}}]_\infty \left( 1 - e^{-\mu t} \right) + [\text{MSI-1*}]_0 e^{-\mu t}$$

$$[\text{mRNA}_{\text{sfGFP}}](t) = \frac{\alpha_y}{\delta} \left( 1 - e^{-\delta t} \right) + [\text{mRNA}_{\text{sfGFP}}]_0 e^{-\delta t}$$

$$[\text{sfGFP}](t) = \beta_y \int_0^t e^{-\mu(t-\tau)} \left( \frac{[\text{mRNA}_{\text{sfGFP}}](\tau)}{1 + \frac{[\text{MSI-1*}](\tau)}{\theta_y}} \right) d\tau + [\text{sfGFP}]_0 e^{-\mu t} \simeq$$
$$\simeq \frac{\alpha_y \beta_y}{\delta} \int_0^t \frac{e^{-\mu(t-\tau)}}{1 + \frac{[\text{MSI-1*}](\tau)}{\theta_y}} d\tau + [\text{sfGFP}]_0 e^{-\mu t},$$

where to perform the approximations $\delta \gg \mu$ is considered (quasi-steady state scenario).

Then, in the steady state, we have

$$\left[\text{mRNA}_{\text{MSI-1*}}\right]_\infty = \frac{\alpha_x}{\delta} \left( \frac{\rho_x + \left(\frac{[\text{Lactose}]}{\theta_x}\right)^{n_x}}{1 + \left(\frac{[\text{Lactose}]}{\theta_x}\right)^{n_x}} \right)$$

$$\left[\text{MSI-1*}\right]_\infty = \frac{\alpha_x \beta_x}{\delta \mu} \left( \frac{\rho_x + \left(\frac{[\text{Lactose}]}{\theta_x}\right)^{n_x}}{1 + \left(\frac{[\text{Lactose}]}{\theta_x}\right)^{n_x}} \right)$$

$$\left[\text{mRNA}_{\text{sfGFP}}\right]_\infty = \frac{\alpha_y}{\delta}$$

$$\left[\text{sfGFP}\right]_\infty = \frac{\alpha_y \beta_y}{\delta \mu} \left( \frac{1}{1 + \frac{\alpha_x \beta_x}{\delta \mu \theta_y} \left( \frac{\rho_x + \left(\frac{[\text{Lactose}]}{\theta_x}\right)^{n_x}}{1 + \left(\frac{[\text{Lactose}]}{\theta_x}\right)^{n_x}} \right)} \right).$$

From the growth curves, we calculated $\mu = 0.8$ h$^{-1}$. Knowing that in *E. coli* the average half-life of mRNA is 5 min (**Bernstein et al., 2002**), we set $\delta = 0.14$ min$^{-1}$. Using our experimental data, the adjusted parameter values are $\frac{\alpha_x \beta_x}{\theta_y} = 13$ h$^{-2}$, $\alpha_y \beta_y = 17$ AU/h$^2$, $\rho_x = 0.075$, $\theta_x = 150$ M, and $n_x = 1.5$ (**Figure 3e–g**).

## Appendix 4

The number of cells ($N$) in a bacterial culture with time can be described by a logistic function (**Peleg and Corradini, 2011**) as

$$N(t) = \frac{N_{\max}}{1 + e^{-\mu(t-\psi)}} \,,$$

where $N_{\max}$ is the maximal capacity of the medium, μ the bacterial growth rate, and $\psi$ the delay of the response (or the time at which the culture reaches half of the capacity).

In our experimental system, the constitutive expression of mScarlet may be used to estimate the total number of cells. Indeed, the absolute red fluorescence level ($\Sigma$mScarlet) may be assumed proportional to $N$. Thus, we may write

$$\Sigma\text{mScarlet}(t) = \frac{\Sigma\text{mScarlet}_{\max}}{1 + e^{-\mu(t-\psi)}}$$

$$\Sigma\text{sfGFP}(t) = \left[\text{sfGFP}\right](t) \cdot \Sigma\text{mScarlet}(t).$$

Using our experimental data, the adjusted parameter values are $\Sigma\text{mScarlet}_{\max}$ = 13.9 AU in the case of no induction, $\Sigma\text{mScarlet}_{\max}$ = 12.5 AU when induced with 1 mM lactose, $\mu$ = 0.8 h$^{-1}$, and $\psi$ = 6.5 h (**Figure 3d**). $\left[\text{sfGFP}\right](t)$ was calculated as described in Appendix 3.

With the time-dependent experimental data in solid media (from LigandTracer), the adjusted parameter values are $\mu$ = 0.0156 min$^{-1}$, $\psi$ = 513 min, $\Sigma\text{mScarlet}_{\max}$ = 1035 AU, and $\left[\text{sfGFP}\right]$ = 6.23 AU in the case of no induction, and $\mu$ = 0.0111 min$^{-1}$, $\psi$ = 630 min, $\Sigma\text{mScarlet}_{\max}$ = 821 AU, and $\left[\text{sfGFP}\right]$ = 2.42 AU when induced with 1 mM IPTG (**Figure 3—figure supplement 2**). In this case, for simplicity, we considered a quasi-steady state scenario, setting constant the sfGFP expression. Moreover, we noticed a delay of about 100 min (= $\nu$) between the mScarlet and sfGFP expressions, so the equation $\Sigma\text{sfGFP}(t) = \left[\text{sfGFP}\right] \cdot \Sigma\text{mScarlet}(t + \nu)$ was used instead to fit the data.

## Appendix 5

List of plasmids used in this work.

| Name | Insert features | Backbone features | Reference |
|---|---|---|---|
| pRM1+ | PLlac:*msi-1\** | KanR, pSC101(E93R) ori | This work |
| pRM0 | void | KanR, pSC101(E93R) ori | This work |
| pRKFR2 | PLlac:*eBFP2* | KanR, pSC101(E93K) ori | *Dolcemascolo et al., 2022* |
| pREP6 | J23119:*sfGFP* (with RNA motif for MSI-1* binding) | CamR, p15A ori | This work |
| pREP6-mut1 | J23119:*sfGFP* (with mutated RNA motif for MSI-1* binding) | CamR, p15A ori | This work |
| pREP6-mut2 | J23119:*sfGFP* (with mutated RNA motif for MSI-1* binding) | CamR, p15A ori | This work |
| pREP6-mut3 | J23119:*sfGFP* (with mutated RNA motif for MSI-1* binding) | CamR, p15A ori | This work |
| pREP6-mut4 | J23119:*sfGFP* (with mutated RNA motif for MSI-1* binding) | CamR, p15A ori | This work |
| pREP6-mut5 | J23119:*sfGFP* (with mutated RNA motif for MSI-1* binding) | CamR, p15A ori | This work |
| pREP7 | J23119:*sfGFP* (with RNA motif for MSI-1* binding and consensus sequences within RBS) | CamR, p15A ori | This work |
| pREP4 | J23119:*sfGFP* (with minimal RNA motif for MSI-1* binding) | CamR, p15A ori | This work |
| pREP4b | J23119:*sfGFP* (with less structured RNA motif for MSI-1* binding) | CamR, p15A ori | This work |
| pREP4b3x | J23119:*sfGFP* (with 3× less structured RNA motifs for MSI-1* binding) | CamR, p15A ori | This work |
| pGio | T7p:*msi-1*$_h$* | KanR, pUC ori | This work |
| pREP6α | PLtet:*sfGFP-mScarlet* (with RNA motif for MSI-1* binding in front of *sfGFP*) | CamR, p15A ori | This work |
| pREP7α | PLtet:*sfGFP-mScarlet* (with RNA motif for MSI-1* binding and consensus sequences within RBS in front of *sfGFP*) | CamR, p15A ori | This work |

# Appendix 6

Nucleotide sequences of the elements used to implement our synthetic gene circuits.

| Name | Sequence |
|---|---|
| PLlac | AATTGTGAGCGGATAACAATTGACATTGTGAGCGGATAACAAGATACTGAGCAC |
| *msi-1\** (codon optimized to *E. coli* from *M. musculus*) | ATGGAAACGGACGCCCCGCAGCCGGGACTGGCCTCTCCTGACTCTCCTCACGACCCATGCAAGATGTTTATTGGTGGACTTTCTTGGCAGACTACTCAGGAGGGTCTTCGTGAATACTTCGGTCAATTTGGCGAAGTGAAAGAGTGTCTTGTGATGCGCGATCCTTTAACCAAGCGTAGTCGCGGATTTGGCTTCGTCACGTTCATGGACCAGGCAGGCGTGGATAAGGTGCTGGCGCAGAGTCGTCACGAATTAGATTCAAAAACGATTGACCCCAAAGTGGCGTTCCCACGTCGCGCCCAACCTAAAATGGTTACTCGTACCAAAAAGATTTTCGTAGGAGGCTTATCCGTAAATACCACGGTAGAAGATGTAAAGCATTACTTCGAACAGTTTGGAAAGGTGGATGATGCAATGCTTATGTTTGATAAGACCACAAACCGTCATCGTGGATTCGGCTTTGTGACCTTTGAATCGGAGGATATCGTTGAGAAGGTCTGCGAAATCCACTTTCATGAAATTAATAACAAATGGTTGAGTGTAAGAAGGCGCAACCGAAAGAAGTCATGTCTCCTTAA |
| J23119 | TTGACAGCTAGCTCAGTCCTAGGTATAATGCTAGC |
| PLtet | TCCCTATCAGTGATAGAGATTGACATCCCTATCAGTGATAGAGATACTGAGCAC |
| *sfGFP* (RNA motif underlined) | ATG<u>GGCAGCGTTAGTTATTTAGTTCGTATGCC</u>AACTAGTCGTAAAGGCGAAGAGCTGTTCACTGGTGTCGTCCCTATTCTGGTGGAACTGGATGGTGATGTCAACGGTCATAAGTTTTCCGTGCGTGGCGAGGGTGAAGGTGACGCAACTAATGGTAAACTGACGCTGAAGTTCATCTGTACTACTGGTAAACTGCCGGTACCTTGGCCGACTCTGGTAACGACGCTGACTTATGGTGTTCAGTGCTTTGCTCGTTATCCGGACCATATGAAGCAGCATGACTTCTTCAAGTCCGCCATGCCGGAAGGCTATGTGCAGGAACGCACGATTTCCTTTAAGGATGACGGCACGTACAAAACGCGTGCGGAAGTGAAATTTGAAGGCGATACCCTGGTAAACCGCATTGAGCTGAAAGGCATTGACTTTAAAGAAGACGGCAATATCCTGGGCCATAAGCTGGAATACAATTTTAACAGCCACAATGTTTACATCACCGCCGATAAACAAAAAAATGGCATTAAAGCGAATTTTAAAATTCGCCACAACGTGGAGGATGGCAGCGTGCAGCTGGCTGATCACTACCAGCAAAACACTCCAATCGGTGATGGTCCTGTTCTGCTGCCCAGACAATCACTATCTGAGCACGCAAAGCGTTCTGTCTAAAGATCCGAACGAGAAACGCGATCATATGGTTCTGCTGGAGTTCGTAACCGCAGCGGGCATCACGCATGGTATGGATGAACTGTACAAATAA |
| RNA motif mutant 1 (A>C substitution) | GGCAGCGTT<u>C</u>GTTATTTAGTTCGTATGCC |
| RNA motif mutant 2 (G>C substitution) | GGCAGCGTTA<u>C</u>TTATTTAGTTCGTATGCC |
| RNA motif mutant 3 (T>C substitution) | GGCAGCGTTAG<u>C</u>TATTTAGTTCGTATGCC |
| RNA motif mutant 4 (G insertion, a nucleotide was deleted downstream to be in frame) | GGCAGCGTTAGTTAT<u>G</u>TTAGTTCGTATGCC |
| RNA motif mutant 5 (two G>C substitutions) | GGCAGCGTTA<u>C</u>TTATTTA<u>C</u>TTCGTATGCC |
| Consensus sequences within RBS | ATCATGTGTTTAGTTAGGAGATTTAGTTA |
| Minimal RNA motif | TTTATAGTTT |
| Less structured RNA motif | GGCGTTAGTTATTTAGTTCGCC |
| 3× less structured RNA motifs | GGCGTTAGTTATTTAGTTCGCCGACGCGTTAGTTTATTTAGTTCGCGATATCCGGTTAGTTATTTAGTTACGG |

*Continued on next page*

*Continued*

| Name | Sequence |
| --- | --- |
| *mScarlet* | ATGGGATCCGTGAGCAAGGGCGAGGCAGTGATCAAGGAGTTCATGCGGTTCAAGGTG<br>CACATGGAGGGCTCCATGAACGGCCACGAGTTCGAGATCGAGGGCGAGGGCGAGGGC<br>CGCCCCTACGAGGGCACCCAGACCGCCAAGCTGAAGGTGACCAAGGGTGGCCCCCTG<br>CCCTTCTCCTGGGACATCCTGTCCCCTCAGTTCATGTACGGCTCCAGGGCCTTCATC<br>AAGCACCCCGCCGACATCCCCGACTACTATAAGCAGTCCTTCCCCGAGGGCTTCAAG<br>TGGGAGCGCGTGATGAACTTCGAGGACGGCGGCGCCGTGACCGTGACCCAGGACACC<br>TCCCTGGAGGACGGCACCCTGATCTACAAGGTGAAGCTCCGCGGCACCAACTTCCCT<br>CCTGACGGCCCCGTAATGCAGAAGAAGACAATGGGCTGGGAAGCGTCCACCGAGCGG<br>TTGTACCCCGAGGACGGCGTGCTGAAGGGCGACATTAAGATGGCCCTGCGCCTGAAG<br>GACGGCGGCCGTTACCTGGCGGACTTCAAGACCACCTACAAGGCCAAGAAGCCCGTG<br>CAGATGCCCGGCGCCTACAACGTCGACCGCAAGTTGGACATCACCTCCCACAACGAG<br>GACTACACCGTGGTGGAACAGTACGAACGCTCCGAGGGCCGCCACTCCACCGGCGGC<br>ATGGACGAGCTGTACAAGTAA |
| *eBFP2* | ATGGTGAGCAAGGGCGAGGAGCTGTTCACCGGGGTGGTGCCCATCCTGGTCGAGCTG<br>GACGGCGACGTAAACGGCCACAAGTTCAGCGTGAGGGGCGAGGGCGAGGGCGATGCC<br>ACCAACGGCAAGCTGACCCTGAAGTTCATCTGCACCACCGGCAAGCTGCCCGTGCCC<br>TGGCCCACCCTCGTGACCACCCTGAGCCACGGCGTGCAGTGCTTCGCCCGCTACCCC<br>GACCACATGAAGCAGCACGACTTCTTCAAGTCCGCCATGCCCGAAGGCTACGTCCAG<br>GAGCGCACCATCTTCTTCAAGGACGACGGCACCTACAAGACCCGCGCCGAGGTGAAG<br>TTCGAGGGCGACACCCTGGTGAACCGCATCGAGCTGAAGGGCGTCGACTTCAAGGAG<br>GACGGCAACATCCTGGGGCACAAGCTGGAGTACAACTTCAACAGCCACAACATCTAT<br>ATCATGGCCGTCAAGCAGAAGAACGGCATCAAGGTGAACTTCAAGATCCGCCACAAC<br>GTGGAGGACGGCAGCGTGCAGCTCGCCGACCACTACCAGCAGAACACCCCCATCGGC<br>GACGGCCCCGTGCTGCTGCCCGACAGCCACTACCTGAGCACCCAGTCCGTGCTGAGC<br>AAAGACCCCAACGAGAAGCGCGATCACATGGTCCTGCTGGAGTTCCGCACCGCCGCC<br>GGGATCACTCTCGGCATGGACGAGCTGTACAAG |

