## [Editor Report · eLife assessment]

This **important** study demonstrates the use of the mammalian Musashi-1 (MSI-1) RNA-binding protein as a tool for regulating gene expression in *Escherichia coli*. The authors provide **convincing** evidence that MSI-1 functions as an effective repressor of translation, and that MSI-1 can be allosterically controlled by oleic acid. This work establishes MSI-1 as a potential tool for synthetic biology applications, and the system developed here can be used for mechanistic studies of MSI-1.

---

## [Referee Report · Joint Public Review]

The authors develop reporter constructs in *E. coli* that are repressed by the mammalian Musashi-1 (MSI-1) RNA-binding protein. Using a set of rigorously controlled experiments, the authors convincingly show that MSI-1 can be directed to control translation, and that translational control by MSI-1 can be modulated allosterically by oleic acid. This is a potentially useful tool for synthetic biologists, with the advantage over transcriptional regulation that one gene in an operon could be targeted. The authors' MSI-1-regulated reporter constructs could also be useful for mechanistic studies of MSI-1.

The authors initial construct design led to only weak regulation by MSI-1, presumably because the MSI-1 binding sites were not suitably positioned to repress translation initiation. A more rationally designed construct led to considerably greater repression. A minor weakness of the paper is that the authors used their initial, weakly regulated construct to assess the effect of MSI-1 binding site mutations and for their mathematical modeling; these experiments would be better suited to the more strongly regulated construct.

---

## [Author Response]

The following is the authors’ response to the original reviews.

Summary of the reviewers’ discussion:

The development of MSI-1 as a post-transcriptional regulator of gene expression in *Escherichia coli* represents a valuable addition to the synthetic biology toolkit. MSI-1 has advantages over transcriptional regulators because it has the potential to target single genes in operons. Allosteric control of MSI-1 by oleic acid increases its versatility.

Authors’ response: We thank the reviewers and editor for this evaluation.

We recommend that authors add experiments to test the mechanism of regulation by MSI-1 or soften their claims about translational regulation. We also recommend that the authors expand their discussion of other natural and synthetic regulatory systems that target translation.

Authors’ response: In this revision, we have added new experimental results from RT-qPCR, bulk fluorometry, and flow cytometry assays to further support our conclusions. We have also enlarged the Introduction and Discussion.

Adding an experiment to quantify the effect of oleic acid with the most strongly regulated reporter construct (i.e., flow cytometry with redesign-3) would substantially increase the impact of the work.

Authors’ response: We have done this experimental quantification (see the new Fig. 5d).

**Reviewer #1 (Public Review):**
The authors develop reporter constructs in *E. coli* where gene expression, presumably translation, is repressed by MSI-1. This is a potentially useful tool for synthetic biologists, with the advantage over transcriptional regulation that one gene in an operon could be targeted. That being said, an important caveat of translational regulation that is not addressed in the manuscript is the potential for downstream effects on RNA stability and/or transcription termination. The authors' MSI-1-regulated reporter constructs could also be useful for mechanistic studies of MSI-1.

Authors’ response: We thank the reviewer for such appreciation of our work. Regarding the potential effects on RNA stability or transcription termination, we would like to highlight our results with the sfGFP-mScarlet bicistron (Fig. 6c), showing the specific regulation of sfGFP by MSI-1* and not of mScarlet. Anyway, for this revision we have conducted an RT-qPCR experiment to quantify the mRNA level of sfGFP to further support our conclusions (see the new Fig. S2).

The author's initial construct design led to only weak regulation by MSI-1, presumably because the MSI-1 binding sites were not suitably positioned to repress translation initiation. A more rationally designed construct led to considerably greater repression. One weakness of the paper is that the authors did not use their redesigned construct that is more strongly repressed to demonstrate allosteric regulation by oleic acid using a comparable assay (e.g., flow cytometry) to that used in other experiments. The potential for allosteric regulation is a major strength of the MSI-1 system, so this is a significant gap. Similarly, the authors use the weakly regulated constructs to assess the effect of MSI-1 binding site mutations and for their mathematical modeling; these experiments would be better suited to the more strongly regulated construct.

Authors’ response: For this revision, we have performed the flow cytometric quantification of the allosteric regulation by oleic acid in the redesigned-3 system (see the new Fig. 5d). Regarding the kinetic study, we focused on the reporter system with just one recognition motif for simplicity. A reporter system with two recognition motifs, thereby recruiting two different proteins, increases the complexity to distill the effect of point mutations.

**Reviewer #1 (Recommendations For The Authors):**
1. Figure 5. Panels c-f look at colonies on plates, with numbers from these data being difficult to compare with either the bulk fluorescence or single-cell fluorescence values shown in other figures. Supplementary Figure 8 shows data for single cells; these data would be more appropriate in Figure 5, with the plate-based data moving to the supplement. Moreover, measuring the effect of oleic acid on the redesign-3 reporter using flow cytometry would assess the impact of oleic acid on the most strongly regulated reporter; this would be the most impactful analysis.

Authors’ response: We have redone Fig. 5 to include flow cytometry data (also for the system implemented with the redesign-3 reporter).

2. Paragraph starting line 438. The authors should briefly discuss the potential for translational repression leading to reduced RNA stability, and in the case of rapid repression that impacts transcription-coupled translation, its impact on Rho-dependent transcription termination. These factors could alter the expression of neighboring genes.

Authors’ response: As we have shown with the RT-qPCR experiment, the mRNA level of the target gene does not change in response to protein binding. We agree that mRNA stability could potentially be changed by using other RNA-targeting proteins. But in our view, a reduction of RNA stability is not a regulation of translation. We have added the following sentence in the Discussion: “The additional use of RNA-binding proteins able to alter mRNA stability might lead to the implementation of more complex circuits at the posttranscriptional level.”

3. Figure 1. It would be informative to include a control where cells have an empty plasmid rather than a plasmid expressing MSI-1, to address leakiness of MSI-1 expression.

Authors’ response: We have constructed a void plasmid as suggested and performed new bulk fluorometry assays. The new Fig. S8 shows the tight control of MSI-1* expression with the PLlac promoter. No apparent leakage is observed.

4. Line 132. Where were the two sequences positioned with respect to each other than the start codon? It would be helpful to show the sequence in Figure 1.

Authors’ response: The precise sequence is shown in the inset of Fig. 1b. The motif is placed just after the start codon.

5. Line 135. The authors envisioned repression mechanism isn't clear from the text, specifically the meaning of "block the progression" and "initial phase". As far as I know, there is no precedent for RNA-binding proteins repressing translation in bacteria by preventing translation elongation. Presumably, repression in the context described here would be due to MSI-1 binding over the ribosome-binding site, although the predicted hairpin may also occlude binding of initiating 30S ribosomes in the absence of MSI-1 binding.

Authors’ response: It is difficult to know the exact mode of action. In page 7, we have rewritten a sentence to have: “In this way, MSI-1* can repress translation by blocking the binding of the ribosome, presumably by imposing a steric hindrance for the 30S ribosomal subunit.”

6. Figure 1e is overly complicated and hence is difficult to interpret. The key result is that mScarlet expression is unchanged as a function of lactose concentration. It is sufficient to show the inset graph as a supplementary figure panel and to conclude that regulation of sfGFP is at a post-transcriptional level. Similarly, the inset in Figure 4b is unnecessary.

Authors’ response: The inset of Fig. 1e shows that the growth rate of the cells is almost constant when lactose varies. A change in growth rate will affect protein expression. The use of a two-reporter system, one regulated translationally and the other not, is instrumental to extract from fluorescence data estimates of transcription and translation rates. Of course, showing that mScarlet expression is almost constant when lactose varies would be sufficient, but we believe that performing a fine treatment of the data helps to better understand the regulatory system from a mathematical and mechanistic point of view. Therefore, despite increasing the complexity of the figure, we prefer to keep the representation of the Crick spaces (following Alon’s terminology, see our ref. 32). We have tried to carefully explain Fig. 1e in the text.

7. Figure 1f and Figure 4c would be easier to interpret as two-dimensional plots.

Authors’ response: We decided to use 3D plots to have more compact representations of the data in the main figures. The accompanying insets show the percentage of cells above the threshold, which helps to understand the regulatory effects. In any case, we have provided the corresponding 2D plots in Fig. S10.

8. I don't think Figure 2e is relevant. The key result is shown in Figure 2f, i.e., the effect of mutations on regulation by MSI-1.

Authors’ response: We agree with the reviewer that the key result is shown in panel f. However, we prefer to keep panel e in Fig. 2 because, even if negative, this result may incite further research. In addition, we avoid the rearrangement of the whole figure.

9. Lines 311-313. Without additional evidence that the mutants are toxic, I suggest removing this text.

Authors’ response: As suggested, we have removed that claim.

**Reviewer #2 (Public Review):**
Summary:Dolcemascolo and colleagues describe the use of the mammalian RNA-binding protein Musashi-1 (MSI-1) to implement translational regulation systems in *E. coli*. They perform detailed in vitro studies of MSI-1 and its binding to different RNA sequences. They provide compelling evidence of the effectiveness of the regulatory system in multiple circuits using different mRNA sequence motifs. They harness allosteric inhibition of MSI-1 by omega-9 monounsaturated fatty acids to demonstrate a fatty-acid-responsive circuit in *E. coli*.Strengths:The experimental results are compelling and the characterization of the binding between MSI-1 and different RNA sequences is thorough and performed via multiple complementary techniques. Several new useful circuit components are demonstrated.

Authors’ response: We thank the reviewer for such appreciation of our work.

Weaknesses:MSI-1 provides 8.6-fold downregulation of sfGFP with an optimized mRNA sequence. In some applications, a larger degree of repression may be required.

Authors’ response: We agree with the reviewer in this point. We expect to conduct further research in the future to optimize the dynamic range of the system. We have added the following sentence in the Discussion: “Further work should be conducted to enhance the fold change of the regulatory module and engineer complex circuits with it.”

**Reviewer #2 (Recommendations For The Authors):**
Overall, I think this paper is very well done and quite thorough. I only have minor suggestions:For Figures 1f and 4c, it is quite hard to interpret the fraction of cells above the threshold with the 3d perspective. It would be clearer to use a more standard 2d plot where the histograms are offset along the y-axis and the threshold is indicated by a vertical line.

Authors’ response: We decided to use 3D plots to have more compact representations of the data in the main figures. The accompanying insets show the percentage of cells above the threshold, which helps to understand the regulatory effects. In any case, we have provided the corresponding 2D plots in Fig. S10.

For Figure 4b, the highlighting of different sequence regions in red3 appears to be offset by one base (e.g. AAU is highlighted rather than AUG).

Authors’ response: This has been corrected.

For line 504, it seems that MSI-1* is used for two different proteins. A different name should be assigned to this 200-residue protein to avoid confusion with the other MSI-1*.

Authors’ response: We now use the term MSI-1h* for the human version of the protein.

The note (Page S12) that A_0 + A_R = alpha/delta only applies in steady-state conditions, which should be stated.

Authors’ response: We have specified that.

It seems that some authors work for the companies that sell some of the instruments/consumables used for the assays, specifically switchSENSE and LigandTracer. This may be something that should be declared under Competing Interests for the paper.

Authors’ response: We are sorry for having missed this point. We have included a Competing Interests section to state that “RAHR and WFV work for Dynamic Biosensors. GPR and JB work for Ridgeview Instruments”.

**Reviewer #3 (Public Review):**
Summary:In this work, the authors co-opt the RRM-binding protein Musashi-1 to act as a translational repressor. The novelty of the work is in the adoption of the allosteric RRM protein Musashi-1 into a translational reporter and the demonstration that RRM proteins, which are ubiquitous in eukaryotic systems, but rare in prokaryotic ones, may act effectively as post-translational regulators in *E. coli*. The extent of repression achieved by the best design presented in this work is not substantially improved compared to other synthetic regulatory schemes developed for *E. coli*, even those that similarly regulate translation (eg. native PP7 repression is approximately 10-fold, Lim et al. J. Biol. Chem. 2001 276:22507-22513). Furthermore, the mechanism of regulation is not established due to missing key experiments. The work would be of broader interest if the allosteric properties of Musashi-1 were more effective in the context of regulation. Unfortunately, the authors do not demonstrate that fatty acids can completely de-repress expression in the experimental system used for most of their assays, nor do they use this ability in their provided application (NIMPLY gate).

Authors’ response: For this revision, we have performed the flow cytometric quantification of the allosteric regulation by oleic acid in the redesigned-3 system, showing substantial de-repression of the system with the biochemical compound. We have redone Fig. 5 and modified the Results section accordingly. Aligned with the reviewers and editor, we believe that this new result helps to improve our manuscript.

Strengths:The first major achievement of this work is the demonstration that a eukaryotic RRM protein may be used to posttranscriptionally regulate expression in bacteria. In my limited literature search, this appears to be the first engineering attempt to design an RBP to directly regulate translation in *E. coli*, although engineered control of translation via other approaches including alterations to RNA structure or via trans-acting sRNAs have been previously described (for review see Vigar and Wieden Biochim Biophys. Acta Gen. Subj. 2017, 1861:3060-3069). Additionally, several viral systems (e.g. MS2 and PP7) have been directly co-opted to work in a similar fashion in the past (utilized recently in Nguyen et al. ACS Synthetic Biol 2022, 11:1710-1718).

Authors’ response: We thank the reviewer for such appreciation of our work.

The second achievement of this work is the demonstration that the allosteric regulation of Musashi-1 binding can be utilized to modulate the regulatory activity. However, the liquid culture demonstration (Suppl. Fig 8) shows that this is not a very effective switch, with de-repressed reporter activity showing substantial change but not approaching un-repressed activity. This effect is stronger when colonies are grown on a solid medium (Fig. 5).

Authors’ response: As we have previously indicated, the flow cytometric quantification of the allosteric regulation by oleic acid in the redesigned-3 system in liquid culture showed substantial de-repression with the biochemical compound. It is now stated in the text the following: “Nevertheless, the system implemented with the redesign-3 reporter displayed a better dynamic behavior in response to lactose and oleic acid. In particular, the percentage of cells in the ON state increased from 0 (with 1 mM lactose) to 71% upon addition of 20 mM oleic acid (Fig. 5d).” This new result helps to improve our manuscript.

Weaknesses:In this work, the authors codon optimize the mouse Musashi-1 coding sequence for expression in *E. coli* and demonstrate using an sfGFP reporter that an engineered Musashi-1 binding site near the translational start site is sufficient to enable a modest reduction in reporter gene expression. The authors postulate that the reduction in expression due to inhibition of ribosome translocation along the transcript (lines 134/135), as an expression of a control transcript (mScarlet) driven by the same promoter (Plac) but without the Musashi-1 recognition site does not demonstrate the same repression. However, the situation could be more complex. Other possibilities include inhibition of translation initiation rather than elongation, as well as accelerated mRNA decay of transcripts that are not actively translated. The authors do not present any measurements of sfGFP mRNA levels.

Authors’ response: In page 7, we have rewritten a sentence to have: “In this way, MSI-1* can repress translation by blocking the binding of the ribosome, presumably by imposing a steric hindrance for the 30S ribosomal subunit.” In addition, for this revision we have conducted an RT-qPCR experiment to quantify the mRNA level of sfGFP to further support our conclusions (see the new Fig. S2). As shown, there is no change in the mRNA level upon inducing the system with lactose.

In subsequent sections of the work, the authors create a series of point mutations to assess RNA-protein binding and assess these via both a sfGFP reporter and in vitro binding assays (switchSENSE). Ultimately, it is difficult to fully rationalize and interpret the behavior of these mutants in the context provided. The authors do identify a relationship between equilibrium constant (1/KD) and fold-repression. However, it is not clear from the narrative why this relationship should exist. Fold-repression is one measure of regulator efficacy, but it is an indirect measure determined from unrepressed and repressed expression. It is not clear why unrepressed expression (in the absence of the protein) is expected to be a function of the equilibrium constant.

Authors’ response: A mathematical derivation from mass action kinetics on why the fold change scales with 1/KD is provided in Note S2. It is the ratio between the unrepressed and repressed expression (i.e., fold change) what scales with 1/KD, but not the expression of a particular state. This kind of relationship has been previously established in the case of transcription regulation [see e.g. Garcia & Phillips, PNAS (2011), our ref. 39]. Our mathematical modeling results expand previous work by providing a single picture from which to analyze transcription and translation regulation.

Subsequent rational redesign of the Musashi-1 binding sequence to produce three alternative designs shows that fold-repression may be improved to approximately 8.6-fold. However, the rationalization of why the best design (red3) achieves this increase based on either the extensive modelling or in vitro measured binding constants is not well articulated. Furthermore, this extent of regulation is approximately that which can be achieved from the PP7 system with its native components (Lim et al. J. Biol. Chem. 2001 276:22507-22513).

Authors’ response: In the case of translation control, the regulation is more challenging because the target is quickly degraded, especially in bacteria (in contrast to transcription control, where the target is stable). This is acknowledged in the manuscript. Even though, it is possible to engineer synthetic circuits with sRNAs or RNA-binding proteins with sufficient dynamic range. We expect to conduct further research in the future to optimize the dynamic range of the system. We have added the following sentence in the Discussion: “Further work should be conducted to enhance the fold change of the regulatory module and engineer complex circuits with it.” Regarding the articulation of the results for the mutants and mathematical model, see our responses in the following questions.

The application provided for this regulator (NIMPLY gate), is not an inherently novel regulatory paradigm, and it does not capitalize on the allosteric properties of Musashi-1, but rather treats Musashi-1 as a non-allosteric component of a regulatory circuit.

Authors’ response: The NIMPLY gate refers to lactose and aTC as inputs. Considering oleic acid as an additional input will lead to a more complex logic. In the last Results section, we wanted to show that the post-transcriptional mechanism engineered with Musashi-1 can be useful specifically regulate a gene within an operon, to implement combinatorial regulation (i.e., coupling transcription and translation control), and to reduce protein expression noise. To these ends, the allosteric ability of the Musashi-1 was not so determinant. In this regard, it would be true that such fine regulatory effects might be achieved as well with non-allosteric RNA-binding proteins, such as MS2CP or PP7CP.

**Reviewer #3 (Recommendations For The Authors):**
1. In the introduction the authors should adequately address the native bacterial mechanisms that allow posttranscriptional regulation in bacteria as well as better discuss previous examples of translational repressors.

Authors’ response: We have added the following paragraph in the Introduction: “Even though bacteria do not appear to exploit proteins to regulate translation in a gene-specific manner, it is worth noting that some bacteriophages do follow this mechanism to modulate their infection cycle. These are the cases, e.g., of the coat proteins of the phages MS2 (infecting *Escherichia coli*) or PP7 (infecting Pseudomonas aeruginosa), which regulate the expression of the cognate phage replicases through protein-RNA interactions [18]. However, one limitation for synthetic biology developments is that such phage proteins are not allosteric. At the post-transcriptional level, bacteria mostly rely on a large palette of cis- and trans-acting non-coding RNAs to either activate or repress protein expression, resulting in the regulation of translation initiation, mRNA stability, or transcription termination, and even allowing sensing small molecules [1,15]. Thus, there should be efforts to replicate this functional versatility with proteins in bacteria.”

2. Given the location of the Musashi-1 binding site in the sfGFP reporter, it may be blocking translation initiation, rather than blocking the progression of the ribosome once attached (line 134/135). The schematic in Fig 1a. is also not overly clear in describing the differences in mechanisms between eukaryotic and prokaryotic systems described in the text.

Authors’ response: In page 7, we have rewritten a sentence to have: “In this way, MSI-1* can repress translation by blocking the binding of the ribosome, presumably by imposing a steric hindrance for the 30S ribosomal subunit.” In page 14, we have added the following sentence: “In this way, MSI-1* can also block the RNA component of the 30S ribosomal subunit.”

3. The authors did not directly examine mRNA levels of their reporter to establish translational regulation. In many cases, inhibition of translation is accompanied by an increased degradation rate in bacterial systems. The authors do not seem to recognize this as a possible amplifier in their system, relying exclusively on normalization via another transcript produced from the same promoter (mScarlet).

Authors’ response: For this revision we have conducted an RT-qPCR experiment to quantify the mRNA level of sfGFP to further support our conclusions (see the new Fig. S2). As shown, there is no change in the mRNA level upon inducing the system with lactose.

4. The results presented for mutations 1-5 are not consistent with the author's models for what is occurring. In particular, mutant 1 displays a reduction in reporter production in the absence of Musashi-1, but the production in the presence does not change from the unaltered sequence. The claim that mutation 1 (in the UAG binding site) results in less binding and ultimately in less regulation is not substantiated since this loss of regulation is due to a reduction in unrepressed expression rather than an increase in expression when Musashi-1 is present.

Authors’ response: We respectfully disagree with this appreciation. In the case of mutant 1, if the Musashi protein recognized the target mRNA with the same affinity as in the original scenario, the red bar would be much lower. Because the Musashi protein hardly recognizes the mutant-1 mRNA, the blue and red bars are quite similar. To clarify this point, we have added the following text in the manuscript: “Despite that mutation substantially reduced sfGFP expression in absence of MSI-1*, the presumed repressed state upon addition of lactose did not change much, suggesting the difficulty of the protein for targeting the mutated mRNA.”

5. Given point 5 above, it is not clear to me why one would expect the 1/KD to be predictive fold-repression in the presence and absence of the repressor. I would rather see the relationship described as predictive in Fig. 2f (fold change vs. 1/KD) rather than the non-linear relationship. It is difficult to qualitatively evaluate the fit quality with the way the data are currently presented.

Authors’ response: Note S2 provides a mathematical derivation from mass action kinetics on why the fold change scales with 1/KD. The R2 value that we provide for the fitting corresponds to the linear regression between fold and 1/KD, as specified in the figure legend. However, we think that the representation of fold vs. KD in log scale is more illustrative in this case.

6. It is not clear what conclusion is determined from the computational modeling, or how this work contributes to the narrative presented. It does not seem like what is learned from these experiments is utilized for novel designs. Furthermore, several of the assumptions within the model may be problematic including the high rate of "elongation leakage" described and the lack of justification for RNA degradation rates utilized.

Authors’ response: The mathematical modeling was performed to rationalize our experimental data. Our idea was more to recapitulate the observed dynamics than to guide the design of new systems. Our model might be exploited to this end in further research, as the reviewer suggests. Besides, elongation leakage is a concept that applies to both transcription and translation regulation systems, and it is not more than the ability of the RNA polymerase or ribosome to elongate even if there is a protein bound to the nucleic acid. This parameter can be set to 0 in the model if appropriate. Moreover, we cite the paper by Bernstein et al., PNAS (2002), our ref. 38, to justify that in *E. coli* the average mRNA half-life is about 5 min (i.e., degradation rate of 0.14 min-1).

7. The data presented in Figure 4 are not presented in a consistent way. While it would be somewhat redundant, including the 0 and 1 mM lactose data for red3 in Figure 4a would be helpful for comparison purposes.

Authors’ response: We have added the requested bar plot in Fig. 4a.

8. The presence of additional Musashi-1 sites upstream of the start codon in red3, and their impact on impact on the fold-repression may support an inhibition of the translation initiation model rather than an inhibition of elongation.

Authors’ response: In page 7, we have rewritten a sentence to have: “In this way, MSI-1* can repress translation by blocking the binding of the ribosome, presumably by imposing a steric hindrance for the 30S ribosomal subunit.” In page 14, we have added the following sentence: “In this way, MSI-1* can also block the RNA component of the 30S ribosomal subunit.”